# UPR<sup>ER</sup>–immunity axis acts as physiological food evaluation system that promotes aversion behavior in sensing low-quality food

UPR$^{ER}$–immunity axis acts as physiological food evaluation system that promotes aversion behavior in sensing low-quality food

**Pengfei Liu, Xinyi Liu, Bin Qi***

Southwest United Graduate School, Yunnan Key Laboratory of Cell Metabolism and Diseases, State Key Laboratory of Conservation and Utilization of Bio-resources in Yunnan, Center for Life Sciences, School of Life Sciences, Yunnan University, Kunming, China

**\*For correspondence:**
qb@ynu.edu.cn

**Competing interest:** The authors declare that no competing interests exist.

**Abstract** To survive in challenging environments, animals must develop a system to assess food quality and adjust their feeding behavior accordingly. However, the mechanisms that regulate this chronic physiological food evaluation system, which monitors specific nutrients from ingested food and influences food-response behavior, are still not fully understood. Here, we established a low-quality food evaluation assay system and found that heat-killed *E. coli* (HK-*E. coli),* a low-sugar food, triggers cellular UPR$^{ER}$ and immune response. This encourages animals to avoid low-quality food. The physiological system for evaluating low-quality food depends on the UPR$^{ER}$ (IRE-1/XBP-1) - Innate immunity (PMK-1/p38 MAPK) axis, particularly its neuronal function, which subsequently regulates feeding behaviors. Moreover, animals can adapt to a low-quality food environment through sugar supplementation, which inhibits the UPR$^{ER}$ -PMK-1 regulated stress response by increasing vitamin C biosynthesis. This study reveals the role of the cellular stress response pathway as physiological food evaluation system for assessing nutritional deficiencies in food, thereby enhancing survival in natural environments.

## eLife assessment

This **valuable** work uses unbiased approaches to discover critical molecules in *C. elegans* and its bacterial food for nutrition sensing and food choice, providing a framework for other studies. The data **convincingly** support their model that *C. elegans* uses UPRER and immune response pathways to evaluate sugar contents in the bacteria to change their behaviors.

## Introduction

Food is essential for the survival, growth, and fitness of all animals. To adapt to fluctuating environments with a wide range of food sources, animals have developed a food evaluation system. This system enables them to identify nutrient-rich food and avoids low-quality or toxic food, thereby maximizing their survival prospects (*Filosa et al., 2016*; *Florsheim et al., 2021*; *McLachlan et al., 2022*). Various sensory neuron evaluation systems in animals has evolved to evaluate food quality through vision (*Avery et al., 2021*; *Melin et al., 2019*), olfactory (*Bargmann, 2006*; *Chalasani et al., 2007*; *Fiala, 2007*; *Ha et al., 2010*; *McLachlan et al., 2022*; *Sengupta et al., 1996*; *Troemel et al., 1997*; *Zhang et al., 2021*) and gustatory senses (*Hukema et al., 2006*; *Ni et al., 2013*; *Scott, 2018*). Besides these sensory systems that facilitate quick feeding decisions, animals may also

**eLife digest** We quickly learn to steer clear of eating the moldy apple, the foul-smelling piece of chicken or the leftovers that taste a little 'off'. This survival instinct is shared across most animal species – even those with extremely simple and limited visual or taste systems, like the tiny worm *Caenorhabditis elegans*. Indeed, assessing the safety and quality of available food items can also rely on cells activating built-in cascades of molecular reactions. However, it remains unclear how these 'cellular stress response programs' actually help guide feeding behaviors.

To better understand this process, Liu et al. conducted a series of experiments using *C. elegans* worms exposed to heat-killed bacteria, which are devoid of many nutrients essential for growth. After initially consuming these bacteria, the worms quickly started to avoid feeding on this type of low-quality food. This suggests that mechanisms occurring after ingestion allowed the worms to adjust their feeding choices.

Further work showed that the consumption of heat-killed bacteria triggered two essential stress response pathways, known as the unfolded protein response and the innate immune response. The activation of these pathways was essential for the animals to be able to change their behavior and avoid the heat-killed bacteria. These biochemical pathways were particularly active in the worms' nerve cells, highlighting the importance of these cells in sensing and reacting to food. Finally, Liu et al. also found that adding sugars like lactose and sucrose to the low-quality food could prevent the activation of the stress response pathways. This result suggests that specific nutrients play a central role in how these worms decide what to eat.

These findings shed light on the complex systems that ensure organisms consume the nutritious food they need to survive. Understanding these processes in worms can provide insights into the broader biological mechanisms that help animals avoid harmful food.

initiate cellular stress response programs to detect nutrition/toxin and trigger food response behaviors (*Jones and Candido, 1999*; *Xie et al., 2022*). This could be one of physiological food quality evaluation systems that monitor the nutritional status of consumed food. However, the signaling events in cellular stress responses involved in evaluating of specific nutrients and the mechanisms that connect these signaling activities to food behaviors are largely unexplored. More specifically, while cellular stress response through UPR^ER (*Richardson et al., 2010*) and PMK-1/p38 MAPK (*Kim et al., 2002*)-dependent immunity in response to pathogens have been extensively studied, the functions of these cellular stress response in sensing and evaluating specific nutrients from food remain unclear.

Vitamin C is an essential micronutrient that cannot be synthesized by humans due to the loss of a key enzyme in the biosynthetic pathway (*Carr and Maggini, 2017*). Animals obtain vitamin C from their diet and possibly also from gut microbes (*Steinert et al., 2020*). Vitamin C is an important physiological antioxidant and a cofactor for a family of biosynthetic and gene regulatory monooxygenase and dioxygenase enzymes. It is also required for the biosynthesis of collagen, L-carnitine, and certain neurotransmitters (*Carr and Frei, 1999*; *Li and Schellhorn, 2007*). Vitamin C has been associated with various human diseases including scurvy, immune defect and cardiovascular disease (*Carr and Maggini, 2017*).

Therefore, in animals, having robust food evaluation systems to detect vitamin C levels could significantly impact their survival in the wild. However, the potential involvement of the cellular stress response pathway in this food evaluation system for sensing and assessing vitamin C remains largely unexplored.

In this study, using the low-quality food evaluation assay system we established (*Qi et al., 2017*), we elucidated the mechanism by which the cellular stress response pathway operates as a physiological food evaluation system. This pathway assesses the deficiency of D-glucose in food and the subsequent vitamin C content in animals through the unfolded protein response (UPR^ER) - innate immunity (PMK-1/p38 MAPK) axis. This mechanism promotes animals to leave low-quality food and is critical for their survival in nature environments.

## Results

### Low-quality food induces stress response in animals

Our previous studies have shown that Heat-killed *E. coli* (HK-*E. coli*), which lacks certain molecules, is considered a low-quality food that is unable to support animal growth (*Qi and Han, 2018*; *Qi et al., 2017*). Moreover, through metabolic-seq analysis, we identified significant changes in a large numbers of derivatives (*Figure 1—figure supplement 1A*, *Supplementary file 1a*), including lipids and their derivatives (*Figure 1—figure supplement 1B*, *Supplementary file 1b*), amino acid and their metabolites (*Figure 1—figure supplement 1C*, *Supplementary file 1c*), as well as coenzymes and vitamins (*Figure 1—figure supplement 1D*, *Supplementary file 1d*). Interestingly, we observed a significant decrease in carboxylic acids and their derivatives (*Figure 1—figure supplement 1E*, *Supplementary file 1e*) in *E. coli* after being heat-killed (*Figure 1—figure supplement 1F*, *Supplementary file 1f*). This suggests that HK-*E. coli* is nutritionally deficient for *C. elegans* when compared to normal *E. coli* food.

Next, we conducted two behavior assays to facilitate the analysis of the food evaluation process in animals by seeding L1 animals in assay plates (*Figure 1A and B*). In the avoidance assay, wild-type animals avoided the HK-*E. coli* OP50 (HK-OP50) food (*Figure 1A*). Interestingly, in the food choice assay, animals initially showed no preference between the two types of food (1–2 hr), but eventually exhibited a preference for high-quality food (Live *E. coli*) up until the 17 hr mark (*Figure 1B*, *Figure 1—figure supplement 1G*). This suggests that worms depart from the HK-*E. coli* after recognizing it as low-quality food source through ingestion.

In order to investigate the pathways in animals that respond to HK-*E. coli*, we performed transcriptomics analysis on worms that were cultured with both HK-*E. coli* and Live *E. coli*. Gene-expression profiling revealed that stress response genes, including those related to biotic stimulus, immune response and response to stress, are up-regulated in animals fed with HK-*E. coli* OP50 (HK-OP50; *Figure 1C*, *Supplementary file 2a*). Among these up-regulated genes, we identified 11 out of 62 of UPR[ER] target genes (*Figure 1D*, *Figure 1—figure supplement 1H* and *Supplementary file 2b*). Additionally, about 50–80% of up-regulated genes overlap with genes responding pathogenic bacteria (*Nakad et al., 2016*; *Sinha et al., 2012*; *Figure 1E*, *Supplementary file 2c*). Consistent with the results of the RNA sequencing (RNA-seq) analysis, the UPR[ER] reporter (*Phsp-4::GFP*) (*Calfon et al., 2002*) and immunity reporter (*Pirg-5::GFP*)(*Bolz et al., 2010*) were strongly induced in intestine (*Figure 1F–G*) and neurons (*Figure 1—figure supplement 2A*) by feeding unfavorable food (HK-*E. coli* OP50), suggesting that UPR[ER] and immune pathways may respond to low-quality food (HK-*E. coli* OP50). As intestinal fluorescence (*Phsp-4::GFP or Pirg-5::GFP*) is easy observation and scoring, the further analyses were done in the intestine.

Moreover, UPR[Mt] reporter (*Phsp-6::GFP*) (*Yoneda et al., 2004*) was weakly induced under HK-*E. coli* feeding condition (*Figure 1—figure supplement 2B*), and starved worm did not induce UPR[ER] and immunity (*Figure 1—figure supplement 2C–D*).

Together, these findings suggest that low-quality food (HK-*E. coli* OP50) triggers a stress response pathway in animals, including UPR[ER] and innate immune pathway. This implies that animals may assess the quality of food through UPR[ER] and innate immune pathway.

### Animals evaluate food quality through UPR[ER]-immune-dependent physiological food quality evaluation system

To determine whether the UPR[ER] and innate immune pathways play a role in evaluating low-quality food, we first examined whether the activation of the UPR[ER] by HK-*E. coli* was dependent on the known signaling components of the UPR[ER] branches, including IRE/XBP-1, PERK/ATF-4 and ATF-6 (*Hetz et al., 2020*; *Ron and Walter, 2007*). We observed no difference in *Phsp-4::GFP* induction with *atf-4* (*Figure 2—figure supplement 1A*) or *atf-6* (*Figure 2—figure supplement 1B*) RNAi-mediated knockdown in animals fed with HK-*E. coli*. However, knockdown of *ire-1/xbp-1* or mutation of *xbp-1* reduced GFP fluorescence (*Figure 2A and B*). Among the 11 differentially expressed UPR[ER] target genes in animals fed with HK-*E. coli* from RNA-seq (*Figure 1D*), 64% of the genes are IRE-mediated genes (*Figure 1—figure supplement 1H*, *Supplementary file 2d*). The mRNA level of IRE-1-mediated splicing of *xbp-1* (*Calfon et al., 2002*) is also induced in animals fed with HK-*E. coli* OP50 (*Figure 2—figure supplement 1C*). However, UPR[ER] is not affected in animals feeding live-*E. coli* by RNAi of *ire-1*,

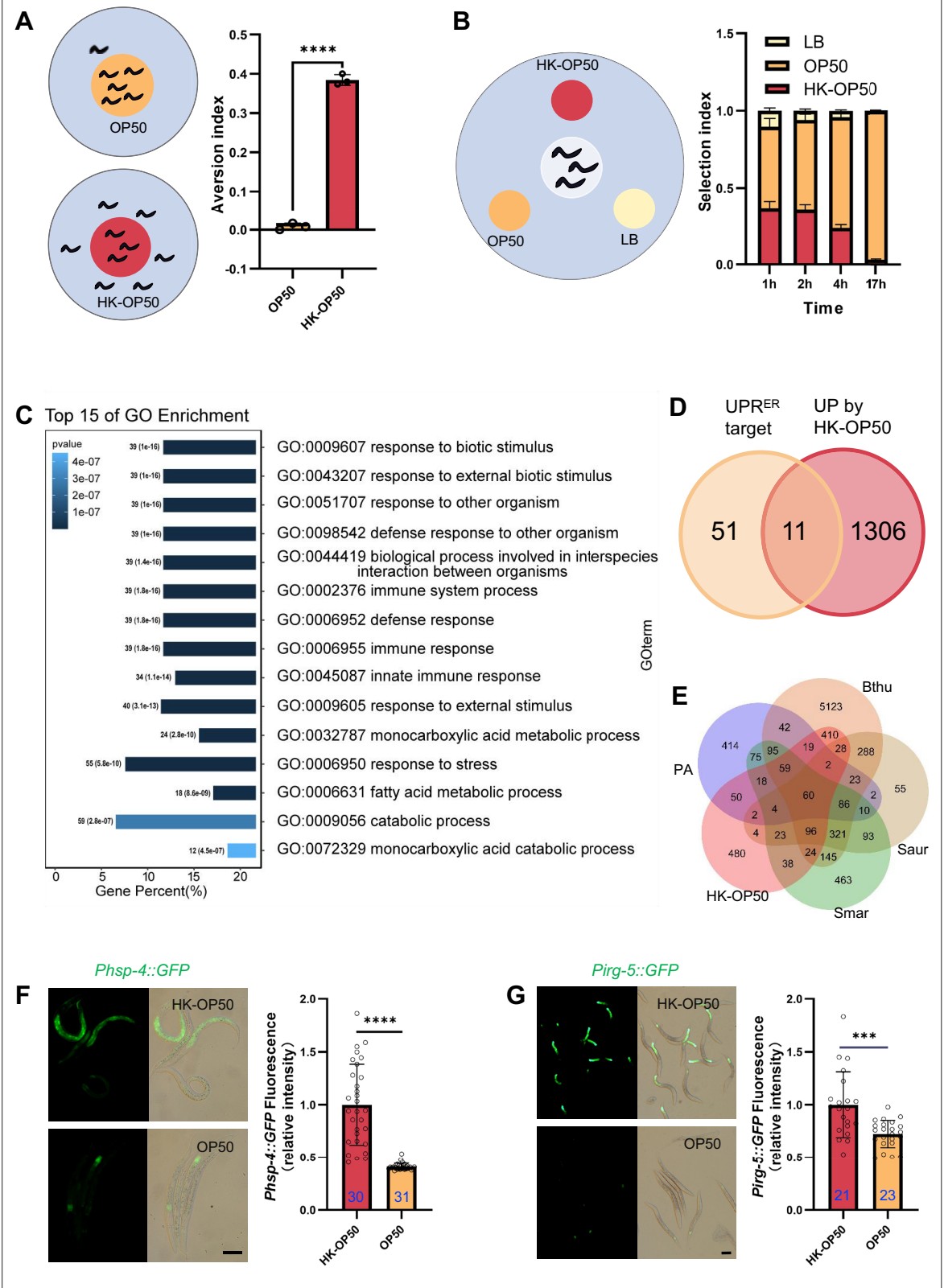

**Figure 1.** The stress response is induced in animals fed low-quality food, HK-*E. coli*. (**A**) Schematic drawing and quantitative data of the food aversion assay. Circles indicate the food spot for live (yellow) and HK-OP50 (red) bacteria, respectively. The animals were scored 16–17 hr after L1 worms were placed on the food spot. Data are represented as mean ± SD from three independent experiments, 79–129 animals/assay. (**B**) Schematic method and quantitative data of the food selection assay. Live (yellow), heat-killed (red) *E. coli* and LB as the buffer for *E. coli* were placed on indicated position.

*Figure 1 continued on next page*

*Figure 1 continued*

Synchronized L1 worms were place in the center spot. The selection index was calculated at the indicated time. Data are represented as mean ± SD from eight independent experiments, 123–792 animals/assay. (**C**) GO enrichment analysis of up-regulated genes in animals fed with HK-*E. coli* vs live *E. coli*. (**D**) Venn diagram showing numbers of UPR^ER target genes and up-regulated genes in animals fed HK-*E. coli*, and their overlap. (**E**) Venn diagram showing numbers of induction genes by four pathogenic bacteria and HK-*E. coli* induced genes, and their overlap. The gene expression data was extracted from published data of animals' infection with *Pseudomonas aeruginosa* (PA) (*Nakad et al., 2016*), *Bacillus thuringiensis* (Bthu) (*Sinha et al., 2012*), *Staphylococcus aureus* (Saur) (*Sinha et al., 2012*), and *Serratia marcescens* (Smar) (*Sinha et al., 2012*). (**F–G**) GFP fluorescence images and bar graph showing that *Phsp-4::GFP* (**F**) and *Pirg-5::GFP* (**G**) were induced in animals fed with HK-*E. coli*. Blue numbers are the number of worms scored from at least three independent experiments. Data are represented as mean ± SD. For all panels, Scale bar shows on indicated figures, 50 µm. * p<0.05, ** p<0.01, *** p<0.001, **** p<0.0001, ns: no significant difference. Precise p values are provided in Raw Data.

The online version of this article includes the following source data and figure supplement(s) for figure 1:

**Source data 1.** Numerical data of *Figure 1A–B and F–G* and *Figure 1—figure supplement 1G*.

**Figure supplement 1.** Food selection assay of animals fed HK-*E. coli* or *E. coli*.

**Figure supplement 2.** Stress response of animals fed HK-*E. coli* or *E. coli*.

*xpb-1*, *atf-4*, and *atf-6* (*Figure 2—figure supplement 1D*). These data suggest that activation of the UPR^ER by low-quality food (HK-*E. coli*) depends on the IRE-1/XBP-1.

To further analyze whether XBP-1-dependent UPR^ER activation is critical for animals to leave low-quality food, we tested food avoidance behavior using *xbp-1* mutant. The results show that *xbp-1* mutants had a significantly decreased likelihood of leaving of HK-*E. coli*, which was rescued by expressing *xbp-1* in neuron rather than intestine (*Figure 2C*). This indicates that XBP-1-dependent UPR^ER activation in neuron is critical for animals to specific evaluate low-quality food (HK-*E. coli*).

We then investigated which innate immune pathway is involved in evaluating low-quality food. First, we analyzed HK-*E. coli*-induced genes from RNA-seq. Among these up-regulated genes, 82 out of 409 of PMK-1-dependent genes (*Fletcher et al., 2019*) were identified (*Figure 2—figure supplement 1E*, *Supplementary file 2c*). Second, we confirmed the induction of several well-known PMK-1 target genes in RNA-seq data (*Foster et al., 2020*; *Figure 2—figure supplement 1F*) and reporter analysis (*Figure 2—figure supplement 1G–H*). Moreover, the induction of *Pirg-5::GFP* was abolished in *pmk-1* knockdown animals fed with HK-*E. coli* (*Figure 2D*). Third, we found that the phosphorylated PMK-1 (p-PMK-1) level was prominently increased in wild-type N2 animals fed HK-*E. coli* compared to feeding *E. coli* OP50 (*Figure 2E*). Finally, *pmk-1* mutants had a decreased likelihood of leaving of HK-*E. coli*, which was rescued by expressing *pmk-1* in neurons rather than intestine (*Figure 2F*). Moreover, *Pirg-5::GFP* is not affected in animals feeding live-*E. coli* by RNAi of *pmk-1* (*Figure 2—figure supplement 1I*).These data suggest that PMK-1 regulated immune pathway evaluates low-quality food, especially the neuronal PMK-1 has a critical function for food quality response.

## XBP-1 and PMK-1 are in the same pathway for evaluating food quality

Next, we explored the connection between UPR^ER (IRE-1/XBP-1) and innate immunity (PMK-1 p38 MAPK) in food quality evaluation. We found that *Pirg-5::GFP* induction (*Figure 2—figure supplements 1I and 2A*) and PMK-1 activation (*Figure 2E*) were decreased in animals with *xbp-1* mutation or knockdown when fed with HK-*E. coli*, suggesting that XBP-1 could regulate PMK-1 under this condition. Additionally, *Phsp-4::GFP* induction under HK-*E. coli* was not affected in animals with *pmk-1* RNAi (*Figure 2—figure supplements 1D and 2B*), indicating that XBP-1-dependent UPR^ER activation is not regulated by PMK-1. Finally, we constructed a double mutant of *xbp-1* and *pmk-1* and found that the food avoidance phenotype of the double mutant was similar to the *pmk-1* mutant (*Figure 2G*), indicating that PMK-1 is downstream of XBP-1 in responding to low-quality food.

We then asked whether UPR^ER (IRE-1/XBP-1) - Innate immunity (PMK-1/p38 MAPK) axis is specific to evaluate low-quality food (HK-*E. coli*). We performed behavior assay in N2, *pmk-1* and *xbp-1* mutant animals by feeding normal *E. coli* food, inedible food (*Saprophytic staphylococci*) (*Geng et al., 2022*) and pathogenic food (*Pseudomonas aeruginosa-PA14*; *Richardson et al., 2010*). We found that N2, *pmk-1*, and *xbp-1* mutant worms did not exhibit avoidance behavior when presented with normal food (OP50). However, both N2 and *xbp-1* mutant worms were able to escape from inedible food (N2 was predominantly found on the border areas of the bacterial lawn and *xbp-1* mutant worms on border and in), *Saprophytic staphylococci*, whereas *pmk-1* mutant worms did not exhibit this avoidance behavior. Notably, N2 and *xbp-1* mutant worms exhibited even more pronounced

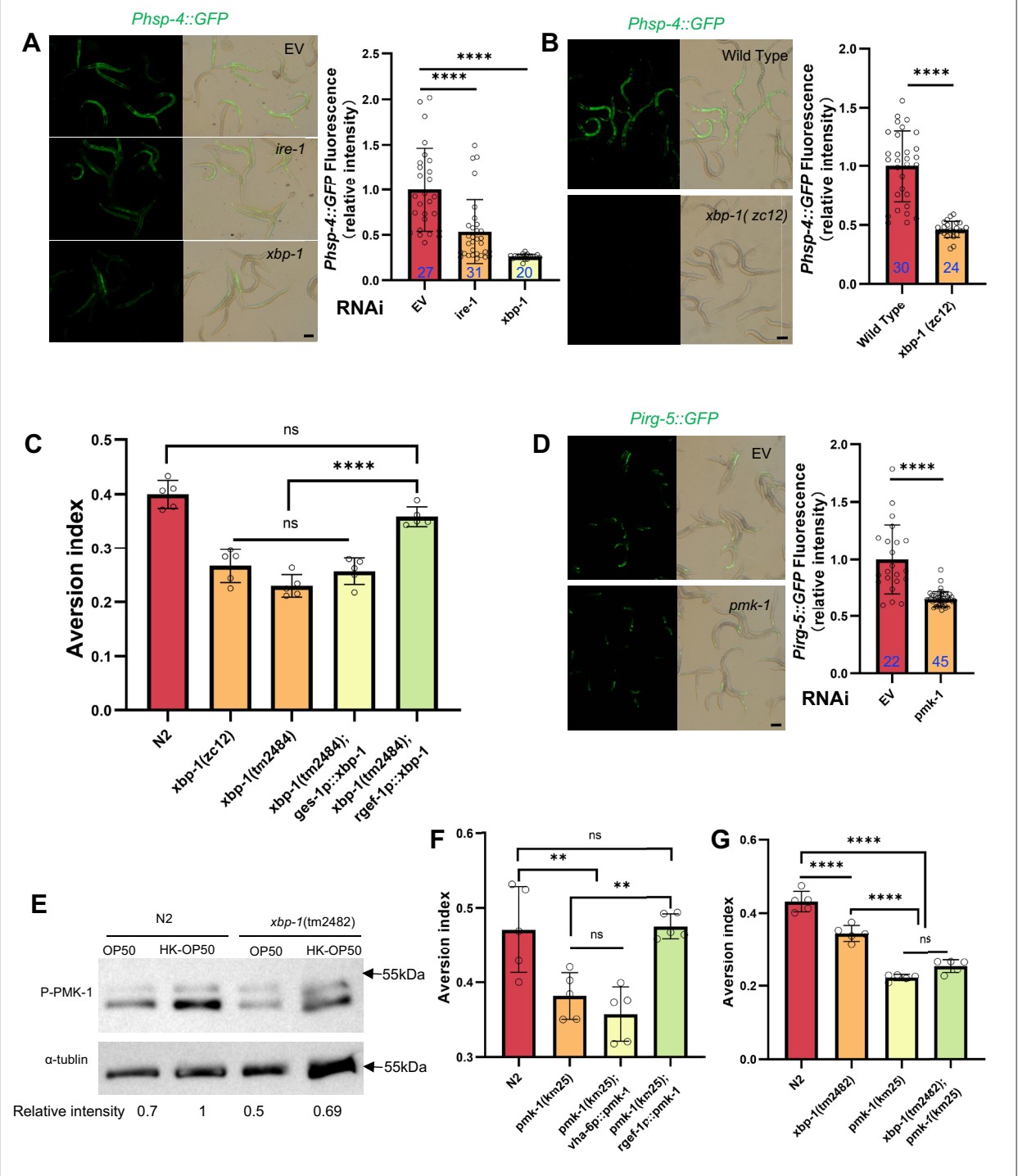

**Figure 2.** Animals evaluate food quality through UPR^ER (*ire-1/xbp-1*) - Innate immunity (*pmk-1* MAPK) axis. (**A–B**) GFP fluorescence images and bar graph showing that HK-*E. coli* induced *Phsp-4::GFP* was decreased in animals with *ire-1* or *xbp-1* RNAi treatment (**A**) or *xbp-1* mutation (**B**). Blue numbers are the number of worms scored from at least three independent experiments. Data are represented as mean ± SD. (**C**) Food aversion assay showing that *xbp-1* mutation eliminated the discrimination against HK-*E. coli*. However, this effect is rescued by expressing *xbp-1* in neurons rather than intestine. Data are represented as mean ± SD from five independent experiments, 156–763 animals/assay. (**D**) GFP fluorescence images and bar graph showing that HK-*E. coli* induced *Pirg-5::GFP* was decreased in animals with *pmk-1* RNAi treatment. Blue numbers are the number of worms scored from at least three independent experiments. Data are represented as mean ± SD. (**E**) Western blot images showing the level of p-PMK-1 in L1 animals (Wild-type N2 and *xbp-1* mutant) fed with OP50 or HK-OP50 for 4 hr. The level of p-PMK-1 is induced in animals fed HK-OP50. (**F**) Food aversion assay showing that *pmk-1* mutation eliminated the discrimination against HK-*E. coli*. However, this effect is rescued by expressing *pmk-1* in neurons rather than intestine.

*Figure 2 continued on next page*

*Figure 2 continued*

Data are represented as mean ± SD from five independent experiments, 168–492 animals/assay. (**G**) Food aversion assay in wild-type, *xbp-1*, *pmk-1* and double mutant. Data are represented as mean ± SD from five independent experiments, 259–490 animals/assay. For all panels, Scale bar shows on indicated figures, 50 µm. * p<0.05, ** p<0.01, *** p<0.001, **** p<0.0001, ns: no significant difference. Precise p values are provided in Raw Data.

The online version of this article includes the following source data and figure supplement(s) for figure 2:

**Source data 1.** Numerical data of *Figure 2A–D and F–G*; *Figure 2—figure supplement 1A–C, F–H*; and *Figure 2—figure supplement 2A–C*.

**Source data 2.** The raw western bolts for *Figure 2E* (labelled).

**Source data 3.** The raw western bolts for *Figure 2E* (unlabelled, uncropped).

**Figure supplement 1.** UPR^ER and innate immunity pathway in animals are critical for evaluating HK-*E. coli*.

**Figure supplement 2.** UPR^ER positively regulates innate immunity pathway in animals.

avoidance behavior when exposed to *Pseudomonas aeruginosa*, whereas *pmk-1* mutant worms were more susceptible to infection by this pathogen (*Figure 2—figure supplement 2C*). These findings suggest that the UPR-Immunity pathway plays a crucial role in helping animals avoid low-quality food (HK-*E. coli*) by triggering an avoidance response. In contrast, the Innate immunity pathway, which is mediated by PMK-1/p38 MAPK, appears to play a key role in evaluating unfavorable food sources, such as HK-*E. coli*, *Saprophytic staphylococci*, and *Pseudomonas aeruginosa*, and helping animals avoid these environments.

## Sugar deficiency in HK-*E. coli* food induces stress response and avoidance behavior in animals

We then investigated which nutrients/metabolites are sensed by animals through the XBP-1-PMK-1 axis for food quality evaluation. First, we hypothesized that the nutrition status is improved in *E. coli* mutant (HK-treatment), which could inhibit UPR^ER and immune response in animals. We established a system for screening the *E. coli* mutant Keio library (*Figure 3—figure supplement 1A*), and identified 20 *E. coli* mutants that did not induce *Phsp-4::GFP* through the UPR^ER reporter (*Phsp-4::GFP*) after three rounds of screening (*Supplementary file 3a*). From these 20 *E. coli* mutants, we identified 9 *E. coli* mutants that did not induce *Pirg-5::GFP* through the immunity reporter (*Pirg-5::GFP*) screening (*Figure 3—figure supplement 1B–C*, *Supplementary file 3a*). Animals fed HK-*yfbR*, which catalyzes carbohydrate derivative metabolic process (*Weiss, 2007*), had a decreased ability to leave food (*Figure 3A*, *Supplementary file 3a*), indicating that HK-*yfbR* may be a higher quality food for animals compared to HK-K12.

Secondly, we performed a metabolomics analysis of different quality food (HK-K12, HK-*yfbR* and Live-K12). We found that the level of 13 metabolites were similar between HK-*yfbR* and Live-K12, but significantly changed in HK-K12 (*Figure 3B*, *Figure 3—figure supplement 2A*, and *Supplementary file 1g*). We also found that genes involved in glycolysis/gluconeogenesis were up-regulated in animals fed with HK-*E. coli* (*Figure 3—figure supplement 2B*), suggesting that glycolysis/gluconeogenesis metabolism is disordered in animals fed with HK-*E. coli*, which may result from changes in sugar/carbohydrate intake. The carbohydrates (D-trehalose, lactose, and D-(+)-sucrose) were also decreased in HK-*E. coli* (*Figure 3B*), suggesting that carbohydrate deficiency may induce stress response and avoidance behavior in animals feeding HK-*E. coli*.

Thirdly, to determine which carbohydrate inhibits stress response in animals, we supplemented each metabolite to HK-*E. coli* and found that only Lactose, and D-(+)-sucrose inhibited HK-*E. coli* induced UPR^ER (*Figure 3C and D*, *Figure 3—figure supplement 2C*). Moreover, we found from our metabolomic data that the sugar level, including lactose, and D-(+)-sucrose, and D-(+)-glucose, was also decreased in HK-*E. coli* (*Figure 3B*, *Supplementary file 1h*). Since lactose and D-(+)-sucrose are hydrolyzed to produce glucose (*Franceus and Desmet, 2020*; *Xing et al., 2019*), we wondered whether glucose also inhibits the stress response in animals. We found that D-(+)-glucose supplementation also inhibited HK-*E. coli* induced UPR^ER (*Figure 3E*), immune response (*Figure 3F and G* and *Figure 3—figure supplement 2D*) and avoidance (*Figure 3H*). Moreover, sugar supplementation did not affect UPR^ER and immunity in normal food (OP50) or starved condition (NGM) (*Figure 3—figure supplement 2E–F*). While sugar effectively inhibits the HK-*E. coli*-induced UPR^ER and immune response, it does not fully suppress it to the extent observed with live-*E. coli* (*Figure 3C–F*). This

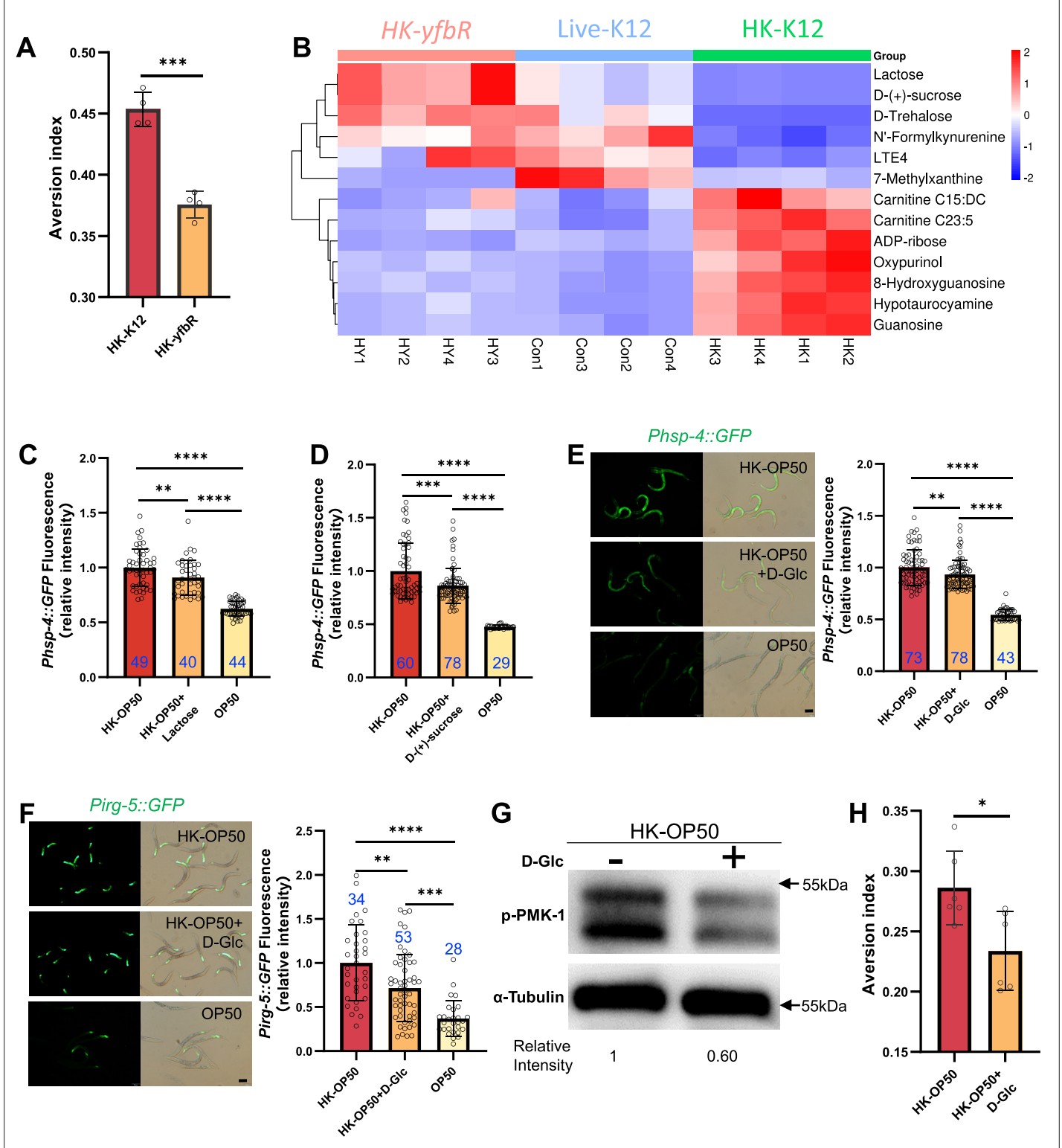

**Figure 3.** HK-*E. coli* is low sugar food, which induce stress response and avoidance behavior in animals. (**A**) Food aversion assay showing that wild-type animals eliminated the discrimination against HK-*E. coli* when *yfbR* is mutated in *E. coli*. Data are represented as mean ± SD four independent experiments, 251–490 animals/assay. (**B**) Heat map showing the 13 differential metabolites from HK-K12, HK-*yfbR*, and K12 in four independent experiments. Color indicates the relative level of each metabolite. (**C–D**) The bar graph showing that HK-*E. coli* induced *Phsp-4::GFP* was decreased in animals with lactose (**C**) or D-(+)-sucrose (**D**) supplementation. Blue numbers are the number of worms scored from at least three independent experiments. Data are represented as mean ± SD. (**E–F**) GFP fluorescence images and bar graph showing that HK-*E. coli* induced *Phsp-4::GFP* (**E**) and

*Figure 3 continued on next page*

Figure 3 continued

*Pirg-5::GFP* (**F**) were decreased in animals with D-(+)-glucose (D-Glc) supplementation. Blue numbers are the number of worms scored from at least three independent experiments. Data are represented as mean ± SD. (**G**) Western blot images showing the level of p-PMK-1 in L1 animals fed HK-*E. coli* with or without D-(+)-glucose (D-Glc) supplementation for 4 hr. The level of p-PMK-1 is decreased in animals fed HK-OP50 +D Glc. (**H**) Food aversion assay showing that wild-type animals eliminated the discrimination against HK-*E. coli* with D-Glc supplementation. Data are represented as mean ± SD six independent experiments, 190–492 animals/assay. For all panels, Scale bar shows on indicated figures, 50 µm. * p<0.05, ** p<0.01, *** p<0.001, **** p<0.0001, ns: no significant difference. Precise p values are provided in Raw Data.

The online version of this article includes the following source data and figure supplement(s) for figure 3:

Source data 1. Numerical data of *Figure 3A, C–F and H*; *Figure 3—figure supplement 1B–C*; and *Figure 3—figure supplement 2C–D, G*.

Source data 2. The raw western bolts for *Figure 3G* (labelled).

Source data 3. The raw western bolts for *Figure 3G* (unlabelled, uncropped).

Figure supplement 1. *E. coli* Keio mutant screening.

Figure supplement 2. Low sugar food, HK-*E. coli*, induce stress response and avoidance behavior in animals.

implies that additional nutrients present in live-*E. coli* might also contribute to the inhibition of UPR$^{ER}$ and immune response.

Previous studies have shown that heat-killed *E. coli* (HK-*E. coli*) is a low-quality food source that cannot support the growth of *C. elegans* larvae (*Qi and Han, 2018*; *Qi et al., 2017*), whereas supplementation with vitamin B2 (VB2) can restore animal growth (*Qi et al., 2017*). Here, we found that sugar deficiency in HK-*E. coli* induces the UPR$^{ER}$-immune response and avoidance behavior in *C. elegans*. Given this, we investigated whether sugar supplementation could promote animal growth when fed HK-*E. coli*. To our surprise, supplementing HK-*E. coli* with carbohydrates (D-Glc, D-GlcA) did not support animal development (*Figure 3—figure supplement 2G*), suggesting that carbohydrates are not sufficient for supporting animal growth on this food source. However, we did find that carbohydrates are critical for inhibiting the UPR$^{ER}$-immune response induced by sugar deficiency in HK-*E. coli*.

Together, these findings suggest that HK-*E. coli* induces a stress response and avoidance behavior in animals, which can be inhibited by D-(+)-glucose supplementation. This implies that animals may evaluate the sugar deficiency from HK-*E. coli* through the activation of UPR$^{ER}$ and immune responses.

## Animals could overcome a low-quality food environment by sugar supplementation through vitamin C biosynthesis

We discovered that D-(+)-glucose supplementation inhibited HK-*E. coli* induced UPR$^{ER}$ (*Figure 3E*), immune response (*Figure 3F and G* and *Figure 3—figure supplement 2D*) and avoidance (*Figure 3H*). Simultaneously, vitamin C (VC), which is synthesized by glucuronate pathway using D-glucose (*Patananan et al., 2015*; *Yabuta et al., 2020*; *Figure 4A*), was found to contribute to neuroprotective (*Moritz et al., 2020*; *Rice, 2000*), immune defense (*Maggini et al., 2007*; *Webb and Villamor, 2007*), and inhibits inflammatory and ER stress (*Luo et al., 2022*; *Su et al., 2019*). This led us to question whether the vitamin C biosynthesis pathway is involved in evaluating low-quality food by using D-glucose.

Firstly, we observed an increase in the vitamin C level in *C. elegans* when fed with HK-*yfbR* (*Figure 4B*), a high-carbohydrate food compared to HK-*E. coli* (*Figure 3B* and *Figure 1—figure supplement 1E*). However, the VC level in bacteria is the same (*Figure 4—figure supplement 1A*). The VC level also increased when D-glucose (*Figure 4—figure supplement 1B*) or D-glucuronate (D-GlcA) was added to HK-*E. coli* (*Figure 4C*), which was abolished by knocking down VC biosynthesis genes (*Figure 4C* and *Figure 4—figure supplement 1B*). This suggests that addition of sugar supplementation promotes VC synthesis in animals fed with HK-*E. coli*.

Secondly, we hypothesized that animals could overcome a low-quality food (HK-*E. coli*) environment by inhibiting the stress response through increasing vitamin C biosynthesis. We found that VC or D-glucuronate (D-GlcA) supplementation inhibits HK-*E. coli* induced UPR$^{ER}$ (*Figure 4D*), immune response including *irg-5/sysm-1* reporter expression (*Figure 4E* and *Figure 4—figure supplement 1C*) and p-PMK-1 (*Figure 4F and G*), as well as food avoidance (*Figure 4H*).

Finally, we asked whether inhibition of stress response and avoidance by sugar supplementation depends on the vitamin C biosynthesis pathway. We found that suppression of HK-*E. coli* induced UPR$^{ER}$ (*Figure 4I*), immune response (*Figure 4J* and *Figure 4—figure supplement 1D*) and food

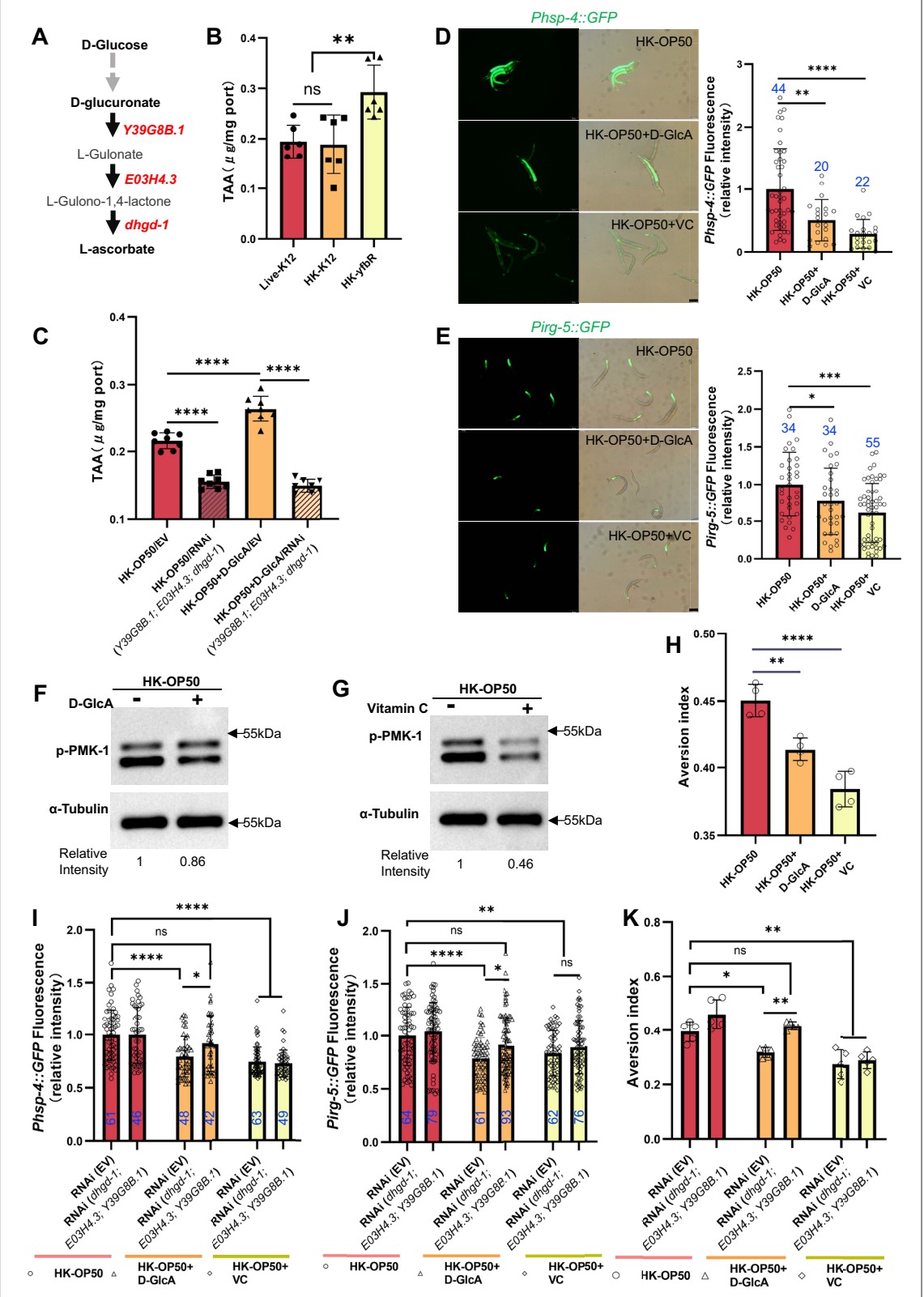

**Figure 4.** Vitamin C biosynthesis pathway is critically involved in evaluating sugar in the food. (**A**) Cartoon illustration of a simplified, Vitamin C biosynthesis pathway in *C. elegans*. The relevant coding genes of enzymes was labeled with red. (**B**) The level of total L-ascorbic acid (TAA), also known as vitamin C, in animals fed with Live-K12, HK-K12, or HK-*yfbR*. Data are represented as mean ± SD from six independent experiments. (**C**) The level of total L-ascorbic acid (TAA) in animals (control or knockdown of Vitamin C biosynthesis genes) fed with HK-*E. coli* with or without D-glucuronate (D-GlcA)

*Figure 4 continued on next page*

*Figure 4 continued*

supplementation. Data are represented as mean ± SD from eight independent experiments. (**D–E**) GFP fluorescence images and bar graph showing that HK-*E. coli* induced *Phsp-4::GFP* (**D**) and *Pirg-5::GFP* (**E**) were decreased in animals with D-GlcA or Vitamin C supplementation. Blue numbers are the number of worms scored from at least three independent experiments. Data are represented as mean ± SD. (**F–G**) Western blot images showing the level of p-PMK-1 in L1 animals fed with HK-*E. coli* with D-GlcA or Vitamin C supplementation for 4 hr. The level of p-PMK-1 is decreased in animals with D-GlcA (**F**) or Vitamin C (**G**) supplementation. (**H**) Food aversion assay showing that wild-type animals eliminated the discrimination against HK-*E. coli* with D-GlcA or Vitamin C supplementation. Data are represented as mean ± SD from four independent experiments, 153–292 animals/assay. (**I–K**) The bar graph showing that suppression of HK-*E. coli* induced *Phsp-4::GFP* (**I**), *Pirg-5::GFP* (**J**) and food avoidance (**K**) by D-GlcA supplementation was abolished in animals with RNAi of VC biosynthesis genes, which was not affect by Vitamin C supplementation. Blue numbers are the number of worms scored from at least three independent experiments and Data are represented as mean ± SD. (**I-J**) Data are represented as mean ± SD from five independent experiments, 252–537 animals/assay. (**K**) For all panels, Scale bar shows on indicated figures, 50 µm. * p<0.05, ** p<0.01, *** p<0.001, **** p<0.0001, ns: no significant difference. Precise p values are provided in Raw Data.

The online version of this article includes the following source data and figure supplement(s) for figure 4:

**Source data 1.** Numerical data of *Figure 4B–E and H–K* and *Figure 4—figure supplement 1A–E*.

**Source data 2.** The raw western bolts for *Figure 4F–G* (labelled).

**Source data 3.** The raw western bolts for *Figure 4F–G* (unlabelled, uncropped).

**Figure supplement 1.** Vitamin C biosynthesis pathway is critical for evaluating low sugar.

---

avoidance (*Figure 4K*) by D-GlcA/sugar supplementation was abolished in animals with RNAi of VC biosynthesis genes. Food selection behavior assays showed that D-GlcA/sugar supplementation increased the preference for heat-killed bacteria, which was also suppressed by knocking down VC biosynthesis genes (*Figure 4—figure supplement 1E*). However, VC supplementation still suppressed the UPR$^{ER}$ (*Figure 4I*), immune response (*Figure 4J* and *Figure 4—figure supplement 1D*) and food avoidance (*Figure 4K*), and increased the food preference (*Figure 4—figure supplement 1E*) in animals with or without RNAi of VC biosynthesis genes. This suggests that VC, as the final metabolite synthesized from D-glucose, is critical for evaluating low-quality food response in animals.

Together, these data indicate that the vitamin C biosynthesis pathway is critical for evaluating whether food is of higher quality and can be eaten by animals. It also suggests that animals could improve their VC levels to adapt to bad food environment.

## Animals evaluate sugar and vitamin C through neuronal XBP-1 and PMK-1

As D-GlcA/sugar and VC supplementation suppressed HK-*E. coli* induced UPR$^{ER}$, immune response and food avoidance behavior, we investigated whether animals evaluate sugar and VC through XBP and PMK-1 dependent pathways. We performed a food selection behavior assay by adding D-Glc, D-GlcA or VC to the NGM, *E. coli* and HK-*E. coli* (*Figure 5A*). The food selection behavior assays revealed that supplementation with D-Glc, D-GlcA, or VC inhibits the animals' choice of sugar or VC on *E. coli*-OP50 feeding conditions (*Figure 5—figure supplement 1A*). This suggests that supplementation with D-Glc, D-GlcA, or VC may alter the metabolites of live bacteria, leading to avoidance by the animals. There was no preference observed on NGM plate (no food condition) supplementation with D-Glc and VC (*Figure 5—figure supplement 1B*), indicating that the intake of sugar or VC alone does not influence animal preference. However, alone D-GlcA could influence worm physiology which induces preference change (*Figure 5—figure supplement 1B*). Interestingly, D-Glc and D-GlcA (*Figure 5B and C*) or VC (*Figure 5D*) supplementation increased the preference for heat-killed bacteria, which was suppressed in *xbp-1* or *pmk-1* mutant animals. However, this preference was also rescued in *xbp-1* or *pmk-1* mutant animals by expressing XBP-1 or PMK-1 in neurons rather than intestine (*Figure 5B–D*), indicating that neuronal XBP-1 and PMK-1 are critical for physiological food elevation system for monitoring the level of sugar and VC under low-quality food condition.

## Discussion

To better survive, animals must evolve a system to recognize and evaluate the quality of their food. This includes the sensory neuron evaluation system for immediate response and feeding decision (*Florsheim et al., 2021*), as well as physiological food evaluation system for chronic response to ingested food. In our previous study, we discovered that the TORC1-ELT-2 pathway, acting as master

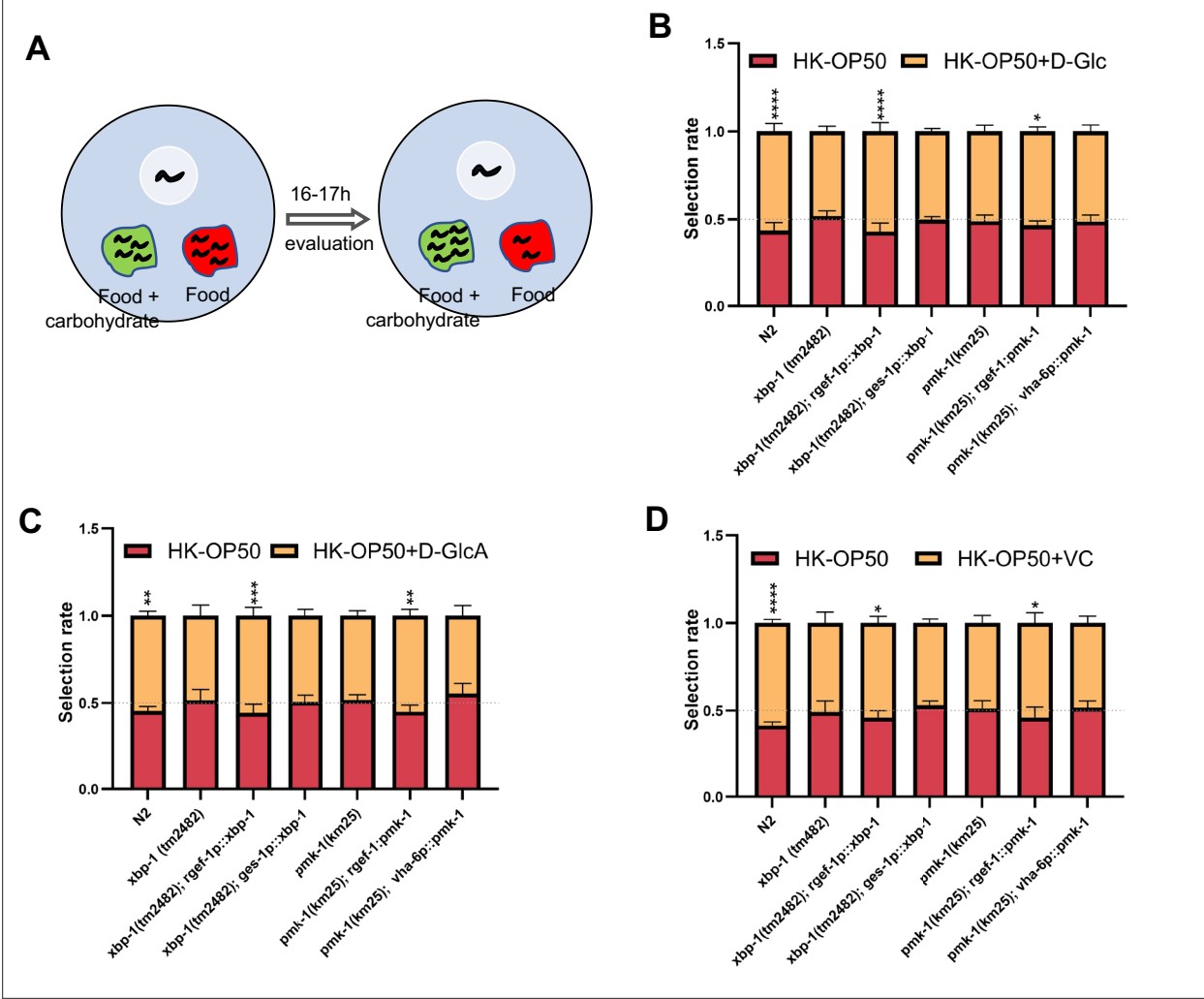

**Figure 5.** Animals evaluate sugar and vitamin C through neuronal XBP-1 and PMK-1. (**A**) Schematic method of the food selection assay. Food (red) and food with carbohydrate (D-Glc, D-GlcA, or VC) supplementation (green) was placed on indicated position. Synchronized L1 worms were then place in plate. After 16-17h, the selection index was calculated. (**B–D**) Food selection assay showing that *xbp-1* or *pmk-1* mutation eliminated the preference of HK-*E. coli* with D-Glc (**B**), D-GlcA (**C**) or Vitamin C (**D**) supplementation, which was rescued in *xbp-1* or *pmk-1* mutant animals by expressing XBP-1 or PMK-1 in neurons rather than intestine. Data are represented as mean ± SD from five independent experiments, 68–647 animals/assay (**B**). Data are represented as mean ± SD from six independent experiments, 83–701 animals/assay (**C**). Data are represented as mean ± SD from six independent experiments, 67–1035 animals/assay (**D**). For all panels, Scale bar shows on indicated figures, 50 μm. * $p<0.05$, ** $p<0.01$, *** $p<0.001$, **** $p<0.0001$, ns: no significant difference. Precise p values are provided in Raw Data.

The online version of this article includes the following source data and figure supplement(s) for figure 5:

**Source data 1.** Numerical data of *Figure 5B–D* and *Figure 5—figure supplement 1A–C*.

**Figure supplement 1.** Food behavior of animals.

**Figure supplement 1—source data 1.** The raw western bolts for *Figure 5—figure supplement 1D* (labelled).

**Figure supplement 1—source data 2.** The raw western bolts for *Figure 5—figure supplement 1D* (unlabelled, uncropped).

regulators in intestine, evaluates vitamin B2 deficiency in low-quality food (HK-*E. coli*) and regulates gut digestive activity to impact animal's food behavior (*Qi et al., 2017*). To further identified the mechanism by which animals evaluate low-quality food (HK-*E. coli*), we performed metabolomics and transcriptomics analyses to identify specific nutrition deficiencies in low-quality food and the cellular response pathways that are involved in food evaluation pathway. This study identified a physiological food evaluation mechanism by which animals recognize food quality through UPR$^{ER}$ (IRE-1/XBP-1) - Innate immunity (PMK-1/p38 MAPK) regulated cellular stress response program in neurons that dictates food avoidance and selection behaviors (*Figure 6*).

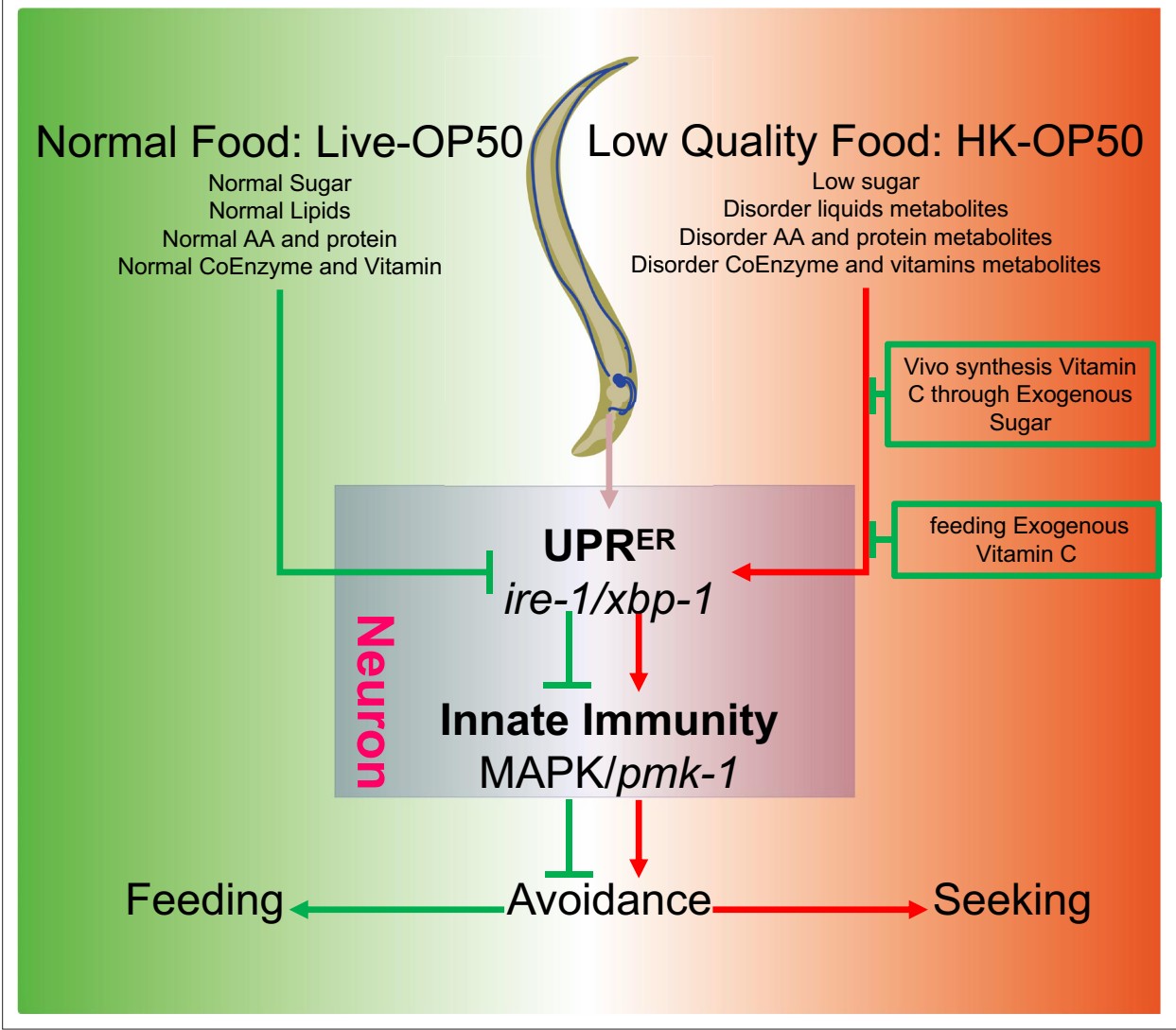

**Figure 6.** Schematic model of physiological food evaluation system in evaluating/sensing sugar and vitamin C through UPR$^{ER}$ (IRE-1/XBP-1) - Innate immunity (PMK-1/p38 MAPK) axis. Vitamin C level is low in animals fed low sugar food, HK-*E. coli*. Sugar and Vitamin C deficiency activate cellular UPR$^{ER}$ and immune response, which promote animals to leave low-quality food and seek better food for survival. This cellular stress regulated physiological food evaluation system depends UPR$^{ER}$ (IRE-1/XBP-1) - Innate immunity (PMK-1/p38 MAPK) axis in neuron.

One of the cellular stress response pathways, the Unfolded Protein Response (UPR$^{ER}$), is activated by various stresses, including infection and nutrition deficiency, which disrupt the homeostasis of the endoplasmic reticulum (ER) (*Hetz et al., 2020*). The activation of UPR$^{ER}$, specifically in the nervous system, has been shown to promote changes in feeding and foraging behavior (*Özbey et al., 2020*). The p38 PMK-1 pathway is also crucial for regulating the expression of secreted innate immune effectors and is essential for survival during infection (*Troemel et al., 2006*). Therefore, these two pathways play a critical role in ensuring animals' survival in changing environments. However, it is still unclear whether UPR$^{ER}$ and innate immunity evaluate food quality under physiological conditions. Our study provides evidence that low-quality food (HK-*E. coli*) activates both UPR$^{ER}$ and p-PMK-1, leading to animals leaving the low-quality food. Previously finding have shown that inhibition of ER function promotes *C. elegans* avoid to the toxic food, which employs the MLK-1/MEK-1/KGB-1 pathway (*Melo and Ruvkun, 2012*). Notably, HK-*E. coli* induced avoidance behavior is independent of the KGB-1 pathway (*Figure 5—figure supplement 1C*). Additionally, our study reveals that neuronal UPR$^{ER}$ and PMK-1 are essential for evaluating low-quality food, suggesting that the nervous system plays a critical role in assessing food quality.

Previous studies have shown that XBP1 deficiency in intestinal epithelial cells leads to IRE1a hyperactivation and increased JNK phosphorylation in the epithelial compartment in vivo (*Kaser et al., 2008*). The IRE1-XBP1 axis has been identified as a critical protective branch of the Unfolded Protein Response (UPR) induced secondary to an innate immune response in the presence of *P. aeruginosa* (*Kaser and Blumberg, 2010*; *Richardson et al., 2010*). The p38 MAPK has also been shown to directly act on the phosphorylation of IRE-1 to promote the stress response (*Guan et al., 2020*; *Qu et al., 2019*). Interestingly, IRE-1 has been found to confer cold resistance independently of XBP-1 by activating JNK-1 MAPK (*Melo and Ruvkun, 2012*). In contrast, our study reveals a new mechanism where the UPR$^{ER}$ (IRE-1/XBP-1) positively regulates Innate Immunity (PMK-1/p38 MAPK) under HK-*E. coli* food conditions, establishing a novel physiological food evaluation system that activates the cellular stress response program.

A previous study has shown that activating innate immunity (PMK-1 MAPK) leads to a reduction in translation (*Weaver et al., 2020*). Our own previous research has also demonstrated that PMK-1 activation causes a shutdown of food digestion in animals (*Geng et al., 2022*), likely to reduce protein translation and cellular metabolism. To investigate this further, we measured the translation level of animals fed with HK-*E. coli* and found that total translation ability is significantly reduced in these animals (*Figure 5—figure supplement 1D*). This finding suggests that activating innate immunity (PMK-1 MAPK) may serve as a mechanism to slow down translation progress, thereby alleviating the pressure on the unfolded protein response (UPR) and preventing excessive UPR$^{ER}$ activation.

Vitamin C (VC) is an important physiological antioxidant and a cofactor for a family of biosynthetic and gene regulatory monooxygenase and dioxygenase enzymes. It is also required for the biosynthesis of collagen, L-carnitine, and certain neurotransmitters (*Carr and Frei, 1999*; *Li and Schellhorn, 2007*). Meanwhile, VC helps animals to protect neuron (*Moritz et al., 2020*; *Rice, 2000*), defend excessive immune (*Maggini et al., 2007*; *Webb and Villamor, 2007*), and inhibit inflammatory and ER stress *Luo et al., 2022*; *Su et al., 2019* in order to better survive. The synthesis of vitamin C (VC) occurs through the glucuronate pathway, utilizing D-glucose as a precursor (*Patananan et al., 2015*; *Yabuta et al., 2020*; *Figure 4A*). This led us to investigate whether the vitamin C biosynthesis pathway is involved in evaluating low-quality food by using D-glucose. In this study, we found that animals feeding live *E. coli*, which should produce more VC, exhibit higher glucose levels. However, our results show that animals maintain similar VC levels when fed ideal food (live *E. coli*) compared to low-quality food (HK-*E. coli*; *Figure 4B*), suggesting that animals do not stimulate VC biosynthesis under favorable food conditions. In contrast, when animals are fed low-quality food (HK-OP50), we found that supplementing D-GlcA in HK-*E. coli* or *E. coli-yfbR* mutation can improve VC levels and inhibit UPR$^{ER}$-immunity (*Figure 4C*). These data indicate that glucose boosts the animal's ability to adapt to unfavorable food environments by increasing VC levels, but not in favorable food conditions.

Unlike the sensory neuron evaluation system, which permits rapid feeding decisions through smell and taste, the cellular stress response as physiological food evaluation system describe here requires a slow and multi-step signal transduction process after the ingestion of food. The disruption of cellular homeostasis by ingested of low-quality or toxic food can activate stress response mechanisms that both increase the cellular ability to withstand and adapt to this disruption of homeostasis and promote behavioral strategies to avoid these conditions and lessen their impact on the organism. These cellular stress response mechanisms include heat shock response, unfolded protein response, oxidative stress response (*Özbey et al., 2021*). Therefore, this slow physiological food evaluation system is an evolutionary adaptation mechanism for detecting nutrition deficiencies in food that was not detected by quick sensory nervous system.

One limitation of our study is the lack of explanation for why HK-*E. coli* activates UPR$^{ER}$ and immunity. We hypothesized that when heat-killed, HK-*E. coli* may lack or contain altered levels of certain metabolites that either activate or inhibit UPR$^{ER}$ and immunity, respectively. Additionally, we speculated that *E. coli* mutants killed by heat may lack metabolites that activate UPR$^{ER}$ and immunity, or conversely, have increased levels of metabolites that inhibit these pathways. Fortunately, our investigation led to the discovery of the *E. coli* mutant *yfbR*, which inhibits UPR$^{ER}$ and immunity by increasing carbohydrates that aid in resisting these stress pathways. Moving forward, we intend to further explore the intricate relationship between HK-*E. coli* and UPR$^{ER}$-immunity. This will be a key focus of our future research efforts.

Collectively, this study uncovers the unexpected function of UPR$^{ER}$ (IRE-1/XBP-1) - Innate immunity (PMK-1/p38 MAPK) as a physiological food evaluation system for evaluating and sensing food quality in animals. It also highlights the utility of the HK-*E. coli* (low-quality food) - *C. elegans* interaction as a means to dissect the mechanism of food evaluation system in assessing food. Most importantly, it reveals that animals are capable of altering their nutrient (Vitamin C) levels through in vivo synthesis or food intake to adapt to a poor food environment when better food choices are not available.

## Materials and methods

### *C. elegans* strains and maintenance

Nematode stocks were maintained on nematode growth medium (NGM) plates seeded with bacteria (*E. coli OP50*) at 20 °C.

(1) The following strains/alleles were obtained from the *Caenorhabditis* Genetics Center (CGC) or as indicated:

N2 Bristol (wild type control strain);
AU78: *agIs219 [T24B8.5p::GFP::unc-54 3' UTR +Pttx-3::GFP::unc-54 3' UTR]*;
SJ4005: *zcIs4 [Phsp-4::GFP]*;
AY101: *acIs101 [F35E12.5p::GFP +rol-6(su1006)]*;
SJ17: *xbp-1 (zc12)*;
KU25: *pmk-1(km25)*;
AY102: *pmk-1(km25) IV; acEx102 [Pvha-6::pmk-1::GFP +rol-6(su1006)]*;
YNU108: *Ex[Prgef-1::pmk-1::GFP; Podr-1::RFP]* (**Geng et al., 2022**);
*xbp-1(tm2482)* (**Richardson et al., 2011**);
KU21 : *kgb-1(km21)*;
AU133: *agIs17 [Pmyo-2::mCherry +Pirg-1::GFP] IV*;
SJ4100: *zcIs13 [Phsp-6::GFP +lin-15(+)]*.

(2) The following strains were constructed by this study:

YNU242: *xbp-1(tm2482); pmk-1(km25)* double mutant was constructed by crossing: *xbp-1(tm2482)* with KU25[*pmk-1(km25)*].
YNU240: *ylfEx149 [xbp-1(tm2482); Prgef-1::xbp-1::GFP; Podr-1::RFP]* transgene strain was constructed by injecting plasmid *Prgef-1::xbp-1::GFP* with *Podr-1::RFP* in *xbp-1(tm2482)* background
YNU241: *ylfEx150 [xbp-1(tm2482); Pges-1::xbp-1::GFP; Podr-1::RFP]* transgene strain was constructed by injecting plasmid P*ges-1:xbp-1::GFP* with *Podr-1::RFP* in *xbp-1(tm2482)* background

### Bacterial strains

*E. coli*-OP50, *Saprophytic staphylococci*, *Pseudomonas aeruginosa*-PA14, *E. coli*-K12 (BW25113), and *E. coli*-K12 mutant were cultured at 37 °C in LB medium. A standard overnight cultured bacteria was then spread onto each Nematode growth media (NGM) plate.

### Culture medium

MGN Medium : Sigma agar: 20 g/L; Bacto Peptone: 2.5 g/L; NaCl: 3 g/L; MgSO4: 0.12 g/L; CaCl: 0.111 g/L; PPB:(KH2PO4 0.8 M; K2HPO4·3H2O 0.2 M) 25 ml/L; Cholesterol: 0.005 g/L.
LB broth: TPYPTONE: 10 g/L; Yeast Extract: 5 g/L; NaCl 5 g/L.

### Method details

#### Generation of transgenes

1. To construct the *C. elegans* plasmid for expression of *xbp-1* in neuron, 3057 bp promoter of *rgef-1* and genomic DNA of *xbp-1* was inserted into the PPD95.77 vector. DNA plasmid mixture containing *Prgef-1::xbp-1::GFP* (25 ng/μl) and *Podr-1::RFP* (25 ng/μl) was injected into the gonads of adult *xbp-1(tm2482)*.

2. To construct the *C. elegans* plasmid for expression of *xbp-1* in intestine, 2549 bp promoter of ges-1 and genomic DNA of *xbp-1* was inserted into the PPD95.77 vector. DNA plasmid mixture containing *Pges-1::xbp-1::GFP* (25 ng/µl) and *Podr-1::RFP (25 ng/µl)* was injected into the gonads of adult *xbp-1(tm2482)*.

## Preparation and feeding of worm food

We followed an established protocol (*Qi and Han, 2018*; *Qi et al., 2017*) to prepare heat-killed (HK) *E. coli*. Briefly, a standard $OD_{600}$=0.5–0.6 of *E. coli* OP50 and *E. coli* K12 grown in LB broth was concentrated to 1/20 vol and was then heat-killed at 80 °C for 180 min. About 150 µl of the heat-killed bacteria was spread onto each 35 mm NGM plate.

## Preparation of HK-*E. coli* + carbohydrate or vitamin C food

1. 100 µl of water, 100 µl of L-ascorbic acid (dissolved in water at a concentration of 100 mg/ml, Sangon Biotech, 100143–0100) or 100 µl of D-glucuronic acid (dissolved in water at a concentration of 100 mg/ml, Adamas, 1102520) was mixed with 500 µl of HK-*E. coli*, then 150 µl of the mixture was spread onto 35 mm NGM plates.
2. 12.5 µl of water or 12.5 µl of D-(+)-glucose (dissolved in water at a concentration of 100 mg/ml, Sangon Biotech, A501991-0500) was mixed with 500 µl of HK-*E. coli*, then 150 µl of the mixture was spread onto 35 mm NGM plates.

## Behavioral assay

### *C. elegans* selection assays

For *C. elegans* to have enough space to evaluate food, we add 18 µl of the sample onto a 35 mm NGM plate. This creates a round lawn with a radius of 5 mm, which occupies about 8% of the total plate area.

$$\frac{a}{b} = \frac{\pi r_1^2}{\pi \gamma_2^2} = \frac{5mm^2}{17.5mm^2} = 8$$

a: the area of bacterial lawn
b: the space of worm life (area of culture dish)

1. 18 µl of heat-killed OP50, live OP50, and LB broth (as the buffer for bacteria) was added into 35 mm NGM plate in an equilateral triangle pattern. Then, synchronized L1 worms were seeded in the center of NGM plate for 16–17 hr at 20 °C (as indicated in *Figure 1B*).
2. 18 µl of heat-killed OP50 and heat-killed OP50 with D-GlcA or vitamin C was added into 35 mm NGM plate in an equilateral triangle pattern, then synchronized L1 worms were seeded on equilateral triangle of NGM plate for 16–17 hr at 20 °C (as indicated in *Figure 5A*).

Here is Selection rate formula:

$$selection\ rate = \frac{\frac{worm\ amount}{one\ lawn\ area}}{\frac{worm\ amount}{lawn1\ area} + \frac{worm\ amount}{lawn2\ area} + \frac{worm\ amount}{lawn3\ area} \cdots}$$

### *C. elegans* aversion assays

18 µl food was spread out the center of NGM plate, then synchronized L1 by bleach solution (NaOH: 1 M, NaClO:4–6%) worms were seeded on center of food for 16–17 hr at 20 °C.

$$\text{Aversion index} = \frac{worm\ amount\ of\ out\ of\ lawn}{worm\ amount\ of\ (out + in)\ lawn}$$

Three to 10 replicates for each condition were performed for each assay, and the experiments were duplicated on different days.

## Analysis of the fluorescence intensity in worms

The synchronized L1 worms carrying either UPR[ER] reporter (*Phsp-4::GFP*) or innate immunity reporter (*Pirg-5::GFP; Psysm-1p::GFP; Pirg-1p::GFP*) were seeded on NGM with indicated food and incubated for 24 hr at 20 °C. For fluorescence imaging, worms were anesthetized with 25 mM levamisole and photographed using either an Olympus BX53 microscope or Olympus MVX10 dissecting microscope equipped with a DP80 camera.

The fluorescence intensity in entire intestinal region was quantified using ImageJ software and normalized to the body area.

## *E. coli* Keio collection screen

The whole Keio *E. coli* single mutant collection (*Baba et al., 2006*) was screened. Mutant bacteria strains, as well as the wild-type control strain BW25113, were cultured in LB medium with 50 µg/ml kanamycin at 37 °C until an $OD_{600}$ of 0.5–0.6 was reached. The bacteria were then heat-killed following our established protocol (*Qi et al., 2017*), and 150 µl of the heat-killed mutant *E. coli* was spread onto 35 mm NGM plates. Synchronized L1 worms carrying UPR[ER] reporter (*Phsp-4::GFP*) were seeded and cultured for 24 hr at 20 °C. The fluorescence was then examined by using an Olympus MVX10 dissecting microscope, progressive screening three times. Next, a 4th screen was performed using immune reporter (*Pirg-5::GFP*) animals fed with HK-E. coli mutants that reduced the *Phsp-4::GFP* fluorescence. Finally, an aversion behavior assay was performed using HK-*E. coli* mutants that both reduced *Phsp-4::GFP* and *Pirg-5::GFP*. HK-*E. coli* mutants that reduced UPR[ER], immune and avoidance behavior were identified through this screening.

## RNAi treatment

RNAi plasmid is delivered in a *E. coli* strain, HT115, from either the MRC RNAi library (*Kamath et al., 2003*) or the ORF-RNAi Library (*Rual et al., 2004*). RNAi plates were prepared by adding IPTG to NGM agar to a final concentration of 1 mM. Overnight *E. coli* cultures (LB broth containing 100 µg/ml ampicillin and 100 uM IPTG) of specific RNAi strains and the control HT115 strain were seeded onto RNAi feeding plates and cultured at room temperature until dry. Synchronized L1 worms were treated RNAi by feeding (Ahringer, Reverse genetics, WormBook 2006) for the first generation and allowed to grow to maturity. The worms were then bleached and hatched in M9 buffer for 18 hr. The synchronized L1 worms were then seeded on the indicated feeding plate.

## Western blot

To measure the level of p-PMK-1, worms (feeding different food for 4 hr) were analyzed by standard western blot methods and probed with anti-p38 (dilution = 1:5000; Cell Signaling, 9212 S), anti-p-p38 (dilution = 1:5000; Cell Signaling, 4511 S) and anti-α-tubulin (dilution = 1:10,000; Sigma T5168) as a loading control.

To measure the level of protein translation, worms (feeding different food for 24 hr) were analyzed by standard western blot methods and probed with anti-Puromycin (dilution = 1:10,000; Sigma-Aldrich, MABE343) and anti-α-tubulin (dilution = 1:10,000; Sigma T5168) as a loading control.

## Total content of ascorbic acid (TAA) assay

The total content of ascorbic acid was measured using the kits (Beijing Biotech-Pack-analytical Scientific Co., Ltd., Beijing, China, BKWB132 http://biotech-pack-analytical.foodmate.net/) according to the manufacturer's protocol. Briefly, L1 worms were seeded on the different feeding assay plate and cultured for 4 hr. The worms were then lysed in ice-cold conditions using lysis buffer. Equal amounts of protein were used for the normalization. Here is formula for getting TAA concentration

$$TAA(ug/mg\ prot) = ((\Delta A - a) \div b) \div (Cpr \times V1) \times D$$

V1 - the volume of supernatant of for experiment
Cpr – the concentration of supernatant protein
D – Dilution ratio of supernatant
a – the intercept of standard curve
b – the slope of standard curve

the standard curve y=0.0611 x+0.0003 for *Figure 4C*; y=0.0258 x+0.0066 for *Figure 4B*, *Figure 4—figure supplement 1A* and *Figure 4—figure supplement 1B*.

## Preparation of samples for RNA sequencing

RNA-seq was done with three biological replicates that were independently generated, collected, and processed. Adult wild type (N2) worms were bleached and then the eggs were incubated in M9 for 18 hr to obtain synchronized L1 worms. L1 worms were cultured in the NGM plate with *HK-E. coli* or *E. coli* for 4 hr at 20 °C. L1 worms were then collected for sequencing.

## RNA sequencing and data processing

For the RNA sequencing assay, cDNA libraries were constructed, and single-end libraries were sequenced using the Illumina platform (Novogene, Beijing, China). HISAT2 (*Mortazavi et al., 2008*) was used to map the clean reads to the reference gene sequence (Species: Caenorhabditis_elegans; Source: NCBI; Reference Genome Version: GCF_000002985.6_WBcel235), and then 'featureCounts' tool in subread software (*Liao et al., 2014*) was used to calculate the gene expression level of each sample. Read counts were inputted into DESeq2 *Love et al., 2014* to calculate differential gene expression and statistical significance. Differentially expressed genes (DEGs) were screened using following criteria: |log2(FoldChange)|>1 & padj ≤ 0.05.

## Preparation of samples for metabolome sequencing

Metabolome-seq of bacterial was done with four biological replicates that were independently generated, collected, and processed. Total of three group *E. coli* sample including: *E. coli K12* (Con), HK-*E. coli K12* (HK), and HK-*E. coli yfbR* mutant (HY). All bacteria were overnight cultured to the same OD ($OD_{600}$=1). *E. coli K12* and *E. coli yfbR* mutant are heat-killed (80, 180 min), *E. coli* K12, HK-*E. coli* K12 and HK-*E. coli yfbR* mutant were then spread out NGM plate for 72 hr at room temperature. Finally, sample was collected into 1.5 ml tube by using sterile cell scraping.

## Metabolome sequencing and data processing

Metabolome were sequenced using the Ultra Performance Liquid Chromatography (UPLC) (ExionLC AD, https://sciex.com.cn/) and Quadrupole-Time of Flight (TripleTOF 6600, AB SCIEX) for Non-targeted; Ultra Performance Liquid Chromatography (UPLC) (ExionLC AD, https://sciex.com.cn/) and Tandem mass spectrometry (MS/MS) (QTRAP, https://sciex.com/) for Broad targeting (Metware, Wuhan, China). Multiple reaction monitoring (MRM) was used to calculate the expression level of each metabolite. Differential metabolites were screened through Fold change ≥2 or Fold change ≤0.5 and VIP ≥1 (Variable Importance in Projection of OPLS-DA model).

## Microscopy

Analysis of fluorescence was performed with an Olympus BX53 microscope, CLSM (Zeiss LSM900), or Olympus MVX10 dissecting with a DP80 camera.

## Quantification and statistical analysis

### Quantification

ImageJ software was used for quantifying fluorescence intensity of $UPR^{ER}$ and Innate immunity reporter. ImageJ software was used for counting the number of worms about selection and aversion behavior.

## Statistical analysis

All statistical analyses were performed in Graphpad prism 8.0. Two-tailed unpaired t test was used for statistical analysis of two groups of samples, one-way or two-way ANOVA was used for statistical analysis of more than two groups of samples. Data are presented as Mean ± SD, and p<0.05 was considered a significant difference, '*' represents p<0.05, '**' represents p<0.01, '***' is represents <0.001, '****' represents p<0.0001, 'ns' represents no significant difference. For all figures, 'n' represents the number of worms scored from at least three independent experiments.

## Acknowledgements

We thank the *Caenorhabditis* Genetics Center (CGC) (funded by NIH P40OD010440) for strains; Dr. Zhao Shan for suggestions. This work was supported by the Ministry of Science and Technology of the People's Republic of China (2019YFA0802100, 2019YFA0803100), the National Natural Science Foundation of China (32170794), Yunnan Provincial Science and Technology Project at Southwest United Graduate School (202302AP370005), Yunnan Applied Basic Research Projects (202201AT070196), Science and Technological Talent Cultivation Plan of Yunnan Province (K264202230211).

## Additional information

### Funding

| Funder | Grant reference number | Author |
| --- | --- | --- |
| Ministry of Science and Technology of the People's Republic of China | 2019YFA0802100 | Bin Qi |
| National Natural Science Foundation of China | 32170794 | Bin Qi |
| Yunnan Provincial Science and Technology Project at Southwest United Graduate School | 202302AP370005 | Bin Qi |
| Yunnan Provincial Science and Technology Department | 202201AT070196 | Bin Qi |
| Science and Technology Talent Cultivation Plan of Yunnan Province | K264202230211 | Bin Qi |
| Ministry of Science and Technology of the People's Republic of China | 2019YFA0803100 | Bin Qi |

The funders had no role in study design, data collection and interpretation, or the decision to submit the work for publication.

### Author contributions

Pengfei Liu, Conceptualization, Resources, Data curation, Software, Formal analysis, Validation, Investigation, Methodology, Writing - original draft, Writing - review and editing; Xinyi Liu, Resources, Investigation, Methodology; Bin Qi, Conceptualization, Resources, Formal analysis, Supervision, Funding acquisition, Validation, Investigation, Visualization, Writing - original draft, Project administration, Writing - review and editing

### Author ORCIDs

Bin Qi ⓘ https://orcid.org/0000-0003-2261-1550

Reviewer #2 (Public Review): https://doi.org/10.7554/eLife.94181.3.sa1
Reviewer #3 (Public Review): https://doi.org/10.7554/eLife.94181.3.sa2
Author response https://doi.org/10.7554/eLife.94181.3.sa3

## Additional files

### Supplementary files

- Supplementary file 1. Metabolism-seq analysis.
- Supplementary file 2. RNA-seq analysis.
- Supplementary file 3. Screening data for *E. coli* mutant keio library.
- Supplementary file 4. Metabolism-seq data of HK-K12, HK-yfbR and K12.

- Supplementary file 5. RNA-seq data of animals fed with HK-*E. coli* OP50 and *E. coli* OP50.
- MDAR checklist

## Data availability

Sequencing data have been deposited in CNCB under accession codes PRJCA028417. All data generated or analysed during this study are included in the manuscript and supporting files; source data files have been provided for all Figures.

The following dataset was generated:

| Author(s) | Year | Dataset title | Dataset URL | Database and Identifier |
|---|---|---|---|---|
| Liu P, Liu X, Qi B | 2024 | RNA-seq fron *C. elegans* fed with *E. coli* OP50 or HK-*E. coli* OP50 | https://ngdc.cncb.ac.cn/bioproject/browse/PRJCA028417 | Genome Sequence Archive, PRJCA028417 |

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
