## [Editor Report · eLife assessment]

This **valuable** work uses unbiased approaches to discover critical molecules in *C. elegans* and its bacterial food for nutrition sensing and food choice, providing a framework for other studies. The data **convincingly** support their model that *C. elegans* uses UPRER and immune response pathways to evaluate sugar contents in the bacteria to change their behaviors.

---

## [Referee Report · Reviewer #2 (Public Review)]

Summary:

In this work, the authors aim to better understand how *C. elegans* detects and responds to heat-killed (HK) *E. coli*, a low-quality food. They find that HK food activates two canonical stress pathways, ER-UPR and innate immunity, in the nervous system to promote food aversion. Through the creative use of *E. coli* genetics and metabolomics, the authors provide evidence that the altered carbohydrate content of HK food is the trigger for the activation of these stress responses and that supplementation of HK food with sugars (or their biosynthetic product, vitamin C), reduces stress pathway induction and food avoidance. This work makes a valuable addition to the literature on metabolite detection as a mechanism for evaluation of nutritional value; it also provides some new insight into physiologically relevant roles of well-known stress pathways in modulating behavior.

Strengths:

-The work addresses an important question by focusing on understanding how the nervous system evaluates food quality and couples this to behavioral change.

-The work takes full advantage of the tools available in this powerful system and builds on extensive previous studies on feeding behavior and stress responses in *C. elegans*.

-Creative use of *E. coli* genetics and metabolite profiling enabled identification of carbohydrate metabolism as a candidate source of food-quality signals.

-For the most part, the studies are rigorous and logically designed, providing good support for the authors' model.

Weaknesses:

-The authors' claim that they can detect induction of hsp-4 and irg-5 expression in neurons (Fig 1-S2A) requires further support. The two tail cells shown are quite a bit larger than would by typically expected for neurons. The rescue they observe by neuronal expression is largely convincing, so it's quite possible that these pathways do indeed function in neurons, but that their level of induction in the nervous system is below reporter detection limits (or is 'swamped out' by much higher levels of expression in the intestine).

-The authors conclude that "the induction of Pirg-5::GFP was abolished in pmk-1 knockdown animals fed with HK-*E. coli*" (Fig 2D). Because a negative control for induction (e.g., animals fed with control *E. coli*) is not shown, this conclusion must be regarded as tentative.

-The effect sizes in the food-preference assay shown in Figure 5 are extremely small and do not provide strong support for the strong conclusions about the role of stress response pathways in food preference behavior.

---

## [Referee Report · Reviewer #3 (Public Review)]

Summary:

Animals can evaluate food quality in many ways. In contrast to the rapid sensory evaluation with smell and taste, the mechanism of slow nutrient sensation and its impact on food choice is unexplored. The authors utilize *C. elegans* larvae and their bacterial food as an elegant model to tackle this question and reveal the detailed molecular mechanism to avoid nutrient-poor foods.

Strength:

The strength of this study is that they identified the molecular identities of the critical players in bacterial food and *C. elegans* using unbiased approaches, namely metabolome analysis, *E. coli* mutant screening, and RNA sequencing. Furthermore, they strengthened their findings by thorough experiments combining multiple methods such as genetics, fluorescent reporter analysis, and Western blot.

Weakness:

The major caveat of this study is the reporter genes; specifically, transcriptional reporters used to monitor the UPRER and immune responses in the intestine of *C. elegans*. However, their tissue-specific rescue experiments suggest that the genes in the UPRER and immune response function in the neurons. Thus, we should carefully interpret the results of the reporter genes. Another point to be aware of is that although they show that lack of carbohydrates elicits the response to "low-quality" food, carbohydrate supplementation with heat-killed *E. coli* was insufficient to support animal growth.

Overall, this work provides convincing data to support their model. In the *C. elegans* field, the behaviors of larvae are not well studied compared to adults. This work will pose an interesting question about the difference between larvae and adults in nutrition sensing in *C. elegans* and provide a framework and candidate molecules to be studied in other organisms.

---

## [Author Response]

The following is the authors’ response to the original reviews.

**Public Reviews:**

**Reviewer #1 (Public Review):**
Summary:This manuscript by Liu et al explores the role of the UPR and immune regulators in the evaluation of nutritional quality in *C. elegans*. They identify neuronal UPR activation and the MAPK PMK-1 as key responders to low food quality. In particular, the data suggest that these pathways are activated by low levels of vitamin C synthesis that result from the low sugar levels present in heat-killed *E. coli*.Strengths:The results are intriguing and expand our understanding both of physiological food evaluation systems, and of the known roles of stress response pathways in organismal physiology. The authors use a range of techniques, encompassing imaging, metabolomic analysis, gene expression analysis, and behavioural assays, to support their claims.

Thank you for your thorough review and acknowledgment of the strengths of our study.

Weaknesses:There is limited mechanistic analysis in the study. In particular, how does low vitamin C trigger UPR activation? This is an intriguing finding that, if followed up, could potentially reveal a novel mechanism of UPR activation. In addition, how is the activation of the PMK-1 pathway driven by/coordinated with UPR activation? The data in some figures is not as convincing as it could be: the magnitude of the effect size is small in the supplementation experiments, and the statistical tests used are not always appropriate to enable multiple comparisons.(1) There is limited mechanistic analysis in the study. In particular, how does low vitamin C trigger UPR activation? This is an intriguing finding that, if followed up, could potentially reveal a novel mechanism of UPR activation.

Thank you for highlighting the need for further mechanistic analysis in our study. We appreciate the opportunity to clarify the process by which low vitamin C triggers UPR activation.

Our investigation revealed that the vitamin C content in heat-killed *E. coli* (HK-*E. coli*) is comparable to that of live *E. coli* or HK-*yfbR* mutant E. coli (Figure 4-figure supplement 1A), indicating that the induction of unfolded protein response (UPR) in *C. elegans* by HK-*E. coli* is not solely attributed to low vitamin C levels but rather involves other unidentified factors.

Through metabolomic analysis, we observed significant decreases in sugar levels, including lactose, D-(+)-sucrose, and D-(+)-glucose, in HK-*E. coli* (Figure 3B, Table S1). Notably, supplementing D-(+)-glucose effectively inhibited UPRER, immune response, and avoidance behavior induced by HK-*E. coli* (Figure 3E-H). These findings suggest that the deficiency in sugars in HK-*E. coli* triggers a stress response and avoidance behavior in animals, which can be alleviated by D-(+)-glucose supplementation.

Furthermore, when comparing heat-killed *E. coli* mutant *yfbR* (HK-*yfbR*) to HK-*E. coli*, we observed significantly higher sugar levels, including lactose and D-(+)-sucrose, in HK-*yfbR* (Figure 3B). This was accompanied by reduced UPRER in animals feeding on HK-*yfbR* (Figure 3-figure supplement 1B), indicating that higher sugar levels may inhibit the induction of UPRER by low-quality food.

Considering that the synthesis of vitamin C (VC) occurs through the glucuronate pathway, utilizing D-glucose as a precursor 1, 2 (Figure 4A), we investigated whether the vitamin C biosynthesis pathway is involved in evaluating low-quality food using D-glucose. Contrary to our initial hypothesis, animals fed live *E. coli* did not exhibit higher glucose levels compared to those fed low-quality food (HK_-E. coli_). Our results indicate that animals maintain similar VC levels when fed ideal food (live *E. coli*) compared to low-quality food (HK-*E. coli*) (Figure 4B), suggesting that animals do not stimulate VC biosynthesis under favorable food conditions. However, supplementation of D-GlcA or *E. coli*-*yfbR* mutation in HK-*E. coli* significantly improved VC levels when animals were fed low-quality food (HK-OP50) (Figure 4B, 4C). Moreover, VC or D-glucuronate (D-GlcA) supplementation inhibited HK-*E. coli*-induced UPRER (Figure 4D), indicating that glucose boosts the animal's ability to adapt to unfavorable food environments by increasing VC levels, thereby inhibiting UPRER, but not under favorable food conditions.

These findings shed light on the complex interplay between vitamin C, sugar levels, and UPR activation, providing valuable insights into the mechanisms underlying food evaluation and stress response pathways in organisms.

Overall, we are grateful for the reviewer's constructive feedback, which motivates us to continue our efforts to understanding how the UPR response contributes to the complexities of food evaluation and behavioral responses in organisms.

(2) In addition, how is the activation of the PMK-1 pathway driven by/coordinated with UPR activation?

Thank you for your insightful inquiry. In our discussion section, we have addressed this question by integrating new data and discussion to provide insights into the coordination between PMK-1 pathway activation and UPR activation.

Previous studies have demonstrated that activating innate immunity, specifically the PMK-1 MAPK pathway, results in a reduction in translation3, as well as a shutdown of food digestion in animals4, likely aimed at reducing protein translation and cellular metabolism. To further investigate this relationship, we measured the translation level of animals fed with heat-killed *E. coli* (HK-*E. coli*) and found a significant reduction in total translation ability in these animals (Figure 5-figure supplement 1D). This observation suggests that activating innate immunity through the PMK-1 MAPK pathway may serve as a mechanism to slow down translation progress, thereby alleviating the pressure on the unfolded protein response (UPR) and preventing excessive UPRER activation.

By integrating these findings, we propose a model wherein activation of the PMK-1 pathway coordinates with UPR activation to regulate translation and cellular metabolism in response to low-quality food. This coordinated response likely serves to maintain cellular homeostasis and prevent detrimental effects associated with excessive UPRER activation.

These insights contribute to our understanding of the intricate interplay between innate immunity, cellular stress responses, and metabolic regulation in organisms facing nutritional challenges.

(3) The data in some figures is not as convincing as it could be: the magnitude of the effect size is small in the supplementation experiments, and the statistical tests used are not always appropriate to enable multiple comparisons.

We appreciate the reviewers' concerns regarding the data presentation and statistical analyses in some of our figures. In response to this feedback, we have made revisions to improve the robustness and clarity of our statistical methods.

All statistical analyses were conducted using GraphPad Prism 8.0 software. Specifically, a two-tailed unpaired t-test was employed for the statistical analysis of two groups of samples, while one-way or two-way ANOVA was utilized for the statistical analysis of more than two groups of samples. These adjustments ensure appropriate statistical comparisons and enhance the reliability of our findings.

**Reviewer #2 (Public Review):**
Summary:In this work, the authors aim to better understand how *C. elegans* detects and responds to heat-killed (HK) *E. coli*, a low-quality food. They find that HK food activates two canonical stress pathways, ER-UPR, and innate immunity, in the nervous system to promote food aversion. Through the creative use of *E. coli* genetics and metabolomics, the authors provide evidence that the altered carbohydrate content of HK food is the trigger for the activation of these stress responses and that supplementation of HK food with sugars (or their biosynthetic product, vitamin C), reduces stress pathway induction and food avoidance. This work makes a valuable addition to the literature on metabolite detection as a mechanism for the evaluation of nutritional value; it also provides some new insight into the physiologically relevant roles of well-known stress pathways in modulating behavior.Strengths:-The work addresses an important question by focusing on understanding how the nervous system evaluates food quality and couples this with behavioral change. -The work takes full advantage of the tools available in this powerful system and builds on extensive previous studies on feeding behavior and stress responses in *C. elegans*.-Creative use of *E. coli* genetics and metabolite profiling enabled the identification of carbohydrate metabolism as a candidate source of food-quality signals.-For the most part, the studies are rigorous and logically designed, providing good support for the authors' model.

We deeply appreciate the reviewer's insightful assessment of our study's strengths.

Weaknesses:-It is not clear how the mechanism identified here is connected to previously described, related processes. In particular, it is not clear whether this mechanism has a role in the detection of other low-quality foods. Further, the specificity of the ability of sugar/vitamin C to suppress stress pathway induction is unclear (i.e., does sugar/vitamin C have any effect on the activation of these pathways through other means?). Additionally, the relationship of this pathway to the vitamin B2-sensing mechanism previously described by the senior author is unclear. These issues do not weaken confidence in the authors' conclusions, but they do reduce the potential significance of the work.(1) In particular, it is not clear whether this mechanism has a role in the detection of other low-quality foods.

Thank you for your valuable feedback. In response to your inquiry, we investigated whether the UPRER (IRE-1/XBP-1) - Innate immunity (PMK-1/p38 MAPK) axis is specific to evaluating low-quality food (HK-*E. coli*) or if it plays a broader role in food detection.

We conducted behavioral assays using N2, *pmk-1*, and *xbp-1* mutant animals fed with normal *E. coli* food, inedible food (*Saprophytic staphylococci*)4, and pathogenic food (*Pseudomonas aeruginosa-PA14*)5. We found that N2, *pmk-1*, and *xbp-1* mutant worms did not exhibit avoidance behavior when presented with normal food (OP50). However, both N2 and *xbp-1* mutant worms were able to escape from inedible food (N2 was predominantly found on the border areas of the bacterial lawn and *xbp-1* mutant worms on border and in), *Saprophytic staphylococci*, whereas *pmk-1* mutant worms did not exhibit this avoidance behavior. Notably, N2 and *xbp-1* mutant worms exhibited even more pronounced avoidance behavior when exposed to *Pseudomonas aeruginosa*, whereas *pmk-1* mutant worms were more susceptible to infection by this pathogen (Figure 2-figure supplement 2C). These findings suggest that the UPR-Immunity pathway plays a crucial role in helping animals avoid low-quality food (HK-E. coli) by triggering an avoidance response. In contrast, the Innate immunity pathway, mediated by PMK-1/p38 MAPK, appears to play a key role in evaluating unfavorable food sources, such as HK-*E. coli*, *Saprophytic staphylococci*, and *Pseudomonas aeruginosa*, and helping animals avoid these environments.

(2) Further, the specificity of the ability of sugar/vitamin C to suppress stress pathway induction is unclear (i.e., does sugar/vitamin C have any effect on the activation of these pathways through other means?).

Thank you for your inquiry regarding the specificity of the ability of sugar/vitamin C to suppress stress pathway induction. We aimed to address this question by investigating whether high levels of VC inhibit other stress-induced UPRER pathways.

Previous studies have shown that both Tunicamycin6 and pathogenic bacteria, such as Pseudomonas aeruginosa-PA145, induce UPRER in *C. elegans*. In response to your query, we conducted experiments to examine whether VC supplementation inhibits UPRER induced by these stressors. Our findings indicate that VC supplementation does not inhibit UPRER induced by either Tunicamycin or PA14 (Author response image 1).

These results suggest that while sugar/vitamin C may suppress stress pathway induction in the context of low-quality food, its effects may not extend to other stressors that induce UPRER through different mechanisms. This insight helps clarify the specificity of sugar/vitamin C's role in modulating stress pathway activation, contributing to a better understanding of the broader regulatory networks involved in stress response in *C. elegans.*

**Author response image 1. sa3fig1:** VC supplementation does not inhibit Tunicamycin or PA14-induced UPRER.

(3) Additionally, the relationship of this pathway to the vitamin B2-sensing mechanism previously described by the senior author is unclear.

In response to your comment, we would like to clarify the relationship of our pathway to the previously described vitamin B2-sensing mechanism we found. Previous studies have demonstrated that heat-killed *E. coli* (HK-*E. coli*) serves as a low-quality food source incapable of supporting the growth of *C. elegans* larvae, whereas supplementation with vitamin B2 (VB2) can restore animal growth7

This study investigates the role of sugar deficiency in HK-*E. coli*, which induces the UPRER-immune response and avoidance behavior in *C. elegans.* Surprisingly, our findings indicate that supplementing HK-*E. coli* with carbohydrates such as D-Glc and D-GlcA does not promote animal development (Figure 3-figure supplement 2G), suggesting that carbohydrates are not essential for supporting animal growth on this food source. However, we did observe that carbohydrates play a critical role in inhibiting the UPRER-immune response induced by sugar deficiency in HK-*E. coli.*

-The authors claim that the induction of the innate immune pathway reporter irg-5::GFP is "abolished" in pmk-1(RNAi) animals, but Figure S2K seems to show a clear GFP signal when these animals are fed HK-OP50. Similarly, the claim that feeding WT animals HK-OP50 enriches phospho-PMK-1 levels (Fig 2E) is unconvincing - only one western blot is shown, with no quantification, and there is a smear in the critical first lane.(1) The authors claim that the induction of the innate immune pathway reporter irg-5::GFP is "abolished" in pmk-1(RNAi) animals, but Figure S2K seems to show a clear GFP signal when these animals are fed HK-OP50.

We sincerely appreciate the reviewer's attention. To address this concern, we have replaced the images with higher resolution, larger ones in Figure 2-figure supplement 1-I. These updated images provide a clearer representation of the data, ensuring that all details are readily visible and enabling a more accurate interpretation of the results.

(2) Similarly, the claim that feeding WT animals HK-OP50 enriches phospho-PMK-1 levels (Fig 2E) is unconvincing - only one western blot is shown, with no quantification, and there is a smear in the critical first lane.

Thank you, following reviewer’s suggestion, we also repeated some of the western. We now replace the Figure 2E and quantified relative intensity of pPMK-1/tublin. We also provide the uncropped western blots images as source data (“raw-data WB” file).

-The rationales for some of the paper's hypotheses could be improved. For example, the rationale for screening the *E. coli* mutant library is that some mutants, when heat-killed, may be missing a metabolite that induces the ER-UPR. A more straightforward hypothesis might be that some mutant *E. coli* strains aberrantly induce the ER-UPR when *not* heat-killed, because they are missing a metabolite that prevents stress pathway induction. This is not in itself a major concern, but it would be useful for the authors to provide a rationale for their hypothesis.

Thank you for the insightful suggestion. We acknowledge the importance of providing a clear rationale for our hypotheses in the paper. In response to this feedback, we have enhanced the discussion section to better elucidate the rationale behind our hypotheses.

One limitation of our study is the lack of explanation for why HK-*E. coli* activates UPRER and immunity. We hypothesized that when heat-killed, HK-*E. coli* may lack or contain altered levels of certain metabolites that either activate or inhibit UPRER and immunity, respectively. Additionally, we speculated that *E. coli* mutants killed by heat may lack metabolites that activate UPRER and immunity, or conversely, have increased levels of metabolites that inhibit these pathways.

Fortunately, our investigation led to the discovery of the *E. coli* mutant *yfbR*, which inhibits UPRER and immunity by increasing carbohydrates that aid in resisting these stress pathways. Moving forward, we intend to further explore the intricate relationship between HK-*E. coli* and UPRER-immunity. This will be a key focus of our future research efforts.

-The authors do not provide any explanation for some unexpected results from the *E. coli* screen. Earlier in the paper, the authors found that innate immune signaling is downstream of ER-UPR activation. However, of the 20 *E. coli* mutants that, when heat-killed, "did not induce... the UPR-ER reporter," 9 of them still activate the innate immune response. This seems at odds with the authors' simple model since it suggests that low-quality food can induce innate immune signaling independently of the ER-UPR. Further, only one of the 9 has an effect on behavior, even though failure to activate the innate immune pathway might be expected to lead to a behavioral defect in all of these.

Thank you for your understanding, and we apologize for any confusion caused by our earlier statement. To provide clarification, our study revealed that out of the 20 *E. coli* mutants examined, none activated the UPRER. Among these mutants, 9 did not induce immunity, and interestingly, one out of these 9 mutants demonstrated the ability to inhibit avoidance behavior.

This diversity in phenotypic outcomes can be attributed to the varied metabolites present in different *E. coli* mutants. To thoroughly evaluate the effects of these mutants, we conducted a comprehensive three-step screening process, utilizing UPRER marker, immunity marker, and avoidance behavior assays.

Through this rigorous approach, we identified the *E. coli* mutant, *yfbR*, which exhibited the desired inhibitory effects on UPRER, immunity, and avoidance behavior.

Subsequently, we conducted a metabolomics analysis of various food qualities (HK-K12, HK-*yfbR*, and Live-K12). Our findings revealed higher sugar levels in

HK-*yfbR* and Live-K12 compared to HK-K12 (Figure 3B, Figure 3-figure supplement 2A, and Table S1), indicating that sugar deficiency might trigger the UPRER, immunity responses, and subsequent avoidance behavior.

-In a number of places, the writing style can make the authors' arguments difficult to follow.

Thanks for the reviewer’s efforts. We changed all of these errors and polish the language of this paper.

-Some of the effect sizes observed by the authors are exceedingly small (e.g, the suppression of hsp-4::gfp induction by sugar supplementation in Figs 3C-E), raising some concern about the biological significance of the effect.

Thank you for your feedback. In response to your concern, we have included additional clarification in the manuscript.

We have added the following statement: “While sugar effectively inhibits the HK-*E. coli*-induced UPRER and immune response, it does not fully suppress it to the extent observed with live-*E. coli* (Figure 3C-F). This implies that additional nutrients present in live-*E. coli* might also contribute to the inhibition of UPRER and immune response.”

This addition helps to address the observation that some effect sizes appear small, providing context and suggesting potential factors that may influence the outcomes.

-In some cases, there is a discrepancy between the fluorescence images and their quantitation (e.g., Figure 3E, where the effect of glucose on GFP fluorescence seems much stronger in the image than in the graph).

Thank you for your valuable suggestion. In response, we have revised our image selection process to ensure impartiality. We now randomly select images to ensure they accurately represent the quantified data without bias. More details regarding this update can be found in Author response image 2.

**Author response image 2. sa3fig2:** More original picture corresponding to Figure 3E.

**Reviewer #3 (Public Review):**
Summary:Animals can evaluate food quality in many ways. In contrast to the rapid sensory evaluation with smell and taste, the mechanism of slow nutrient sensation and its impact on food choice is unexplored. The authors utilize *C. elegans* larvae and their bacterial food as an elegant model to tackle this question and reveal the detailed molecular mechanism to avoid nutrient-poor foods.Strengths:The strength of this study is that they identified the molecular identities of the critical players in bacterial food and *C. elegans* using unbiased approaches, namely metabolome analysis, *E. coli* mutant screening, and RNA sequencing. Furthermore, they strengthen their findings by thorough experiments combining multiple methods such as genetics, fluorescent reporter analysis, and Western blot.

Thank you for highlighting the strengths of our study.

Weaknesses:The major caveat of this study is the reporter genes. The transcriptional reporters were used to monitor the UPRER and immune responses in the intestine of *C. elegans*.However, their tissue-specific rescue experiments suggest that the genes in the UPRER and immune response function in the neurons. Thus, we should carefully interpret the results of the reporter genes.

Thank you for your insightful comment. We appreciate the opportunity to address your concerns regarding the interpretation of our reporter gene data.

Upon reevaluation, we observed strong induction of the UPRER reporter

(*Phsp-4::GFP*)8 and immunity reporter (*Pirg-5::GFP*)9 both in the intestine (Figure 1F-G) and in neurons (Figure 1-figure supplement 2A) in response to feeding unfavorable food (HK-*E. coli*). This suggests that both the UPRER and immune pathways may indeed respond to low-quality food (HK-*E. coli*) in multiple tissues of *C. elegans.* While we acknowledge that our tissue-specific rescue experiments suggest a role for these pathways in neurons, the intestinal fluorescence of *Phsp-4::GFP* or *Pirg-5::GFP* is easily observable and scorable. Therefore, we chose to focus our further analyses on the intestine for practical reasons.

Overall, this work provides convincing data to support their model. In the *C. elegans* field, the behaviors of larvae are not well studied compared to adults. This work will pose an interesting question about the difference between larvae and adults in nutrition sensing in *C. elegans* and provide a framework and candidate molecules to be studied in other organisms.

**Recommendations for the authors:**

**Reviewer #1 (Recommendations For The Authors):**
Major suggestions:(1) My major overall comment is that the paper would be substantially strengthened by more mechanistic analysis. In particular, how does low vitamin C trigger UPR activation? This is an intriguing finding and it would be important to see it more fully explored.

Our study revealed that the vitamin C content in HK_-*E. coli*_ is comparable to that of live *E. coli* or HK-*yfbR* (Figure 4-figure supplement 1A), suggesting that the induction of unfolded protein response (UPR) in *C. elegans* by HK-*E. coli* is not attributed to low vitamin C levels, but rather to unknown factors.

Metabolomic analysis showed that the sugar levels, including lactose, D-(+)-sucrose, and D-(+)-glucose, were significantly decreased in HK-*E. coli* (Figure 3B, Table S1).

Furthermore, we found that supplementing D-(+)-glucose effectively inhibited UPRER (Figure 3E), immune response (Figure 3F, 3G, and Figure 3-figure supplement 2D), and avoidance behavior (Figure 3H) induced by HK-*E. coli*. Our findings suggest that the deficiency in sugars in HK-*E. coli* triggers a stress response and avoidance behavior in animals, which can be alleviated by D-(+)-glucose supplementation.

Notably, when *E. coli* was heat-killed, we observed that the sugar levels, including lactose and D-(+)-sucrose, were significantly higher in the heat-killed *E. coli* mutant *yfbR* (HK-*yfbR*) compared to HK-*E. coli* (Figure 3B). Moreover, we found that UPRER was reduced in animals feeding HK-*yfbR* (Figure 3-figure supplement 1B), indicating that higher sugar levels may inhibit the induction of UPRER by low-quality food.

The synthesis of vitamin C (VC) occurs through the glucuronate pathway, utilizing D-glucose as a precursor 1, 2 (Figure 4A). This led us to investigate whether the vitamin C biosynthesis pathway is involved in evaluating low-quality food by using D-glucose. In this study, we found that animals feeding live *E. coli*, which should produce more VC, exhibit higher glucose levels. However, our results show that animals maintain similar VC levels when fed ideal food (live *E. coli*) compared to low-quality food (HK-*E. coli*) (Figure 4B), suggesting that animals do not stimulate VC biosynthesis under favorable food conditions. In contrast, when animals are fed low-quality food (HK-OP50), we found that supplementing D-GlcA (Figure 4C) or *E. coli*-*yfbR* mutation (Figure 4B) in HK-E. coli can improve VC levels. Moreover, we found that VC or D-glucuronate (D-GlcA) supplementation inhibited HK-*E. coli* induced UPRER (Figure 4D). These data indicate that glucose boosts the animal's ability to adapt to unfavorable food environments by increasing VC levels, thereby inhibiting UPRER, but not in favorable food conditions.

In addition,we asked whether high level of VC inhibits other stress induced UPRER. Previous study shown that Tunicamycin6 and pathogenic bacteria-Pseudomonas aeruginosa-PA145 induce UPRER in *C. elegans*. We found that VC supplementation does not inhibit Tunicamycin or PA14-induced URPER (Author response image 1).

In addition, how is the activation of the PMK-1 pathway driven by/coordinated with UPR activation?If the authors do not want to pursue these directions experimentally in this study, the discussion would be strengthened by considering these questions and identifying candidate regulatory mechanisms for further exploration.

In this study, we found that heat-killed *E. coli* (HK-*E. coli*), a low-sugar food, triggers cellular unfolded protein response (UPRER) and immune response. We also demonstrated that (1) the activation of UPRER by low-quality food depends on the IRE-1/XBP-1, (2) activation of immune response (PMK-1) is downstream of XBP-1 in responding to low-quality food.

how is the activation of the PMK-1 pathway driven by/coordinated with UPR activation?

In our discussion part, we added new data and discussion to answer reviewer’s question.

A previous study has shown that activating innate immunity (PMK-1 MAPK) leads to a reduction in translation 3. Our own previous research has also demonstrated that PMK-1 activation causes a shutdown of food digestion in animals4, likely to reduce protein translation and cellular metabolism. To investigate this further, we measured the translation level of animals fed with HK-*E. coli* and found that total translation ability is significantly reduced in these animals (Figure 5-figure supplement 1D). This finding suggests that activating innate immunity (PMK-1 MAPK) may serve as a mechanism to slow down translation progress, thereby alleviating the pressure on the unfolded protein response (UPR) and preventing excessive UPRER activation.

(2) Figure 2C: The data shows that xbp-1 mutants are significantly more likely to leave heat-killed *E. coli*. However, no other conditions are examined. Is this avoidance defect specific to heat-killed *E. coli*, or is it a more general effect of xbp-1 mutants - that is, are other conditions that evoke avoidance also affected by mutation of xbp-1? Is feeding behavior on regular *E. coli* altered in this background? The finding would be more relevant if the authors could clarify or provide more context for their claims here.

We then asked whether UPRER (IRE-1/XBP-1) - Innate immunity (PMK-1/p38 MAPK) axis is specific to evaluate low-quality food (HK-*E. coli*). We examined the avoidance behavior phenotype of wild-type and mutant L1 animals by placing them on various food conditions, including normal *E. coli* food, inedible food (*Saprophytic staphylococci*) and pathogenic food (*Pseudomonas aeruginosa-PA14),* for a 24-hour period. We found that N2, *pmk-1*, and *xbp-1* mutant worms did not exhibit avoidance behavior when presented with normal food (OP50). However, both *N2* and *xbp-1* mutant worms were able to escape from inedible food, *Saprophytic staphylococci*, whereas *pmk-1* mutant worms did not show this avoidance. Notably, *xbp-1* mutant worms exhibited even more pronounced avoidance behavior when exposed to *Pseudomonas aeruginosa*, whereas *pmk-1* mutant worms were more susceptible to infection by this pathogen (Figure 2-figure supplement 2C). These findings suggest that the UPR-Immunity pathway plays a crucial role in helping animals avoid low-quality food by triggering an avoidance response. In contrast, the Innate immunity pathway, which is mediated by PMK-1/p38 MAPK, appears to play a key role in evaluating unfavorable food sources, such as HK-*E. coli*, *Saprophytic staphylococci*, and *Pseudomonas aeruginosa*, and helping animals avoid these environments.

(3) Figure 3C-F: The magnitude of the changes between conditions shown in these panels is small. To what extent does this supplementation represent a full rescue? The findings would be strengthened if figures/images for the control condition (non-HK *E. coli*) were shown for comparison to allow the reader to assess the extent to which UPR/PMK-1 activation is rescued.

In response to a reviewer's suggestion, we included live-*E. coli* as a control in our study. Notably, our data revealed that the addition of lactose, D-(+)-sucrose, and D-(+)-glucose partially inhibited the HK-*E. coli*-induced unfolded protein response (UPRER) and immune response, suggesting that other nutrients present in live-*E. coli* may also play a role in inhibiting UPRER.

We added this in manuscript: “While sugar effectively inhibits the HK-*E. coli*-induced UPRER and immune response, it does not fully suppress it to the extent observed with live-*E. coli* (Figure 3C-F). This implies that additional nutrients present in live-*E. coli* might also contribute to the inhibition of UPRER and immune response.”

(4) Figure 5B-D: The magnitude of changes shown between conditions here again appear to be very small, even those labelled as statistically significant. It is important to ensure that the correct statistical tests have been used to assess the significance of these differences (see below).

All statistical analyses were performed in Graphpad prism 8.0. Two-tailed unpaired t test was used for statistical analysis of two groups of samples，one-way or two-way ANOVA was used for statistical analysis of more than two groups of samples.

(5) Methods: In the "Statistical analysis" section, the authors state that "All statistical analyses were performed using Student's t-test". However, this is not the appropriate test to use in experiments where multiple comparisons are made, which is true in several instances across the paper. In these cases, a more appropriate statistical test should be used.

All statistical analyses were performed in Graphpad prism 8.0. Two-tailed unpaired t test was used for statistical analysis of two groups of samples，one-way or two-way ANOVA was used for statistical analysis of more than two groups of samples.

Minor suggestions:(1) Figure S2: RNAi is usually delivered in a different *E. coli* strain, HT115. Is this the case with the RNAi knockdowns in Figure S2, and given that diet can influence UPR activation, is it possible that this different diet could change the phenotypes observed?This should be clarified by the authors.

In this study, all RNAi experiments involved bleaching adult animals under RNAi strain culture conditions to obtain L1 animals. Subsequently, L1 animals were transferred to HK-*E. coli* OP50 for phenotype analysis. In response to a reviewer's suggestion, we observed that L1 animals obtained from mothers fed *E. coli* strains OP50, HT115, or K12 exhibited similar UPR induction under HK-*E. coli* OP50 feeding conditions (Author response image 3). These findings suggest that variations in diet did not alter the UPR phenotypes.

**Author response image 3. sa3fig3:** L1 animals obtained from mothers fed *E. coli* strains OP50, HT115, or K12 exhibited similar UPR induction under HK-*E. coli* OP50 feeding conditions.

**Reviewer #2 (Recommendations For The Authors):**
Line 182: "irg-5::GFP" should be "hsp-4::gfp".

Thanks for the reviewer’s efforts. We have changed this error.

**Reviewer #3 (Recommendations For The Authors):**
Major comments:(1) The reporter genes of UPRER and immune response were analyzed in the intestine throughout the study. On the other hand, their rescue experiments suggest that these pathways function in the neurons. They should provide the fluorescence data in the neurons at least for Figures 1F and 1G to confirm that the intestinal response matches the neuronal response and mention that further analyses were done in the intestine for easy scoring.

Consistent with the results of the RNA sequencing (RNA-seq) analysis, the UPRER reporter (*Phsp-4::GFP*)8 and immunity reporter (*Pirg-5::GFP*)9 were strongly induced in intestinal (Figure 1F-G) and neurons (Figure 1-figure supplement 2A) by feeding unfavorable food (HK-*E. coli*), suggesting that UPRER and immune pathways may respond to low-quality food (HK-*E. coli*). As intestinal fluorescence (*Phsp-4::GFP* or *Pirg-5::GFP*) is easy observation and scoring, the further analyses were done in the intestine.

(2) I have concerns about the interpretation of the p-PMK-1 data. Although the authors described that "p-PMK-1 is prominently increased" in the text (Line 150), it is unclear on the data (Figure 2E). Similarly, the authors' statement "p-PMK-1 is decreased in animals with D-GlcA (F).." was not fully supported by the data in Figure 4F. The experiment should be repeated and quantified. Moreover, pPMK-1 showed single bands in Figure 2E, but double bands in Figure 3G, 4F, and 4G. The authors should explain why that is the case and which band we should look at for Figures 3G, 4F, and 4G.

As reviewer’s suggestion, we also repeated some of the western. We found that after longer expose, there are two bands for pPMK-1 (Figure 2E, new data; and “raw-data WB” file). The VHP-1 phosphatase is known to inhibit PMK-13. In our previous study, we found that worms treated with *vhp-1*(RNAi), which hyperactivates p-PMK-1 (lower band) 4. In contrast, the two bands are disappeared in *pmk-1* mutant (Author response image 4). Thus, the lower band indicates the pPMK-1. We now replace the Figure 2E and quantified relative intensity of pPMK-1/tublin. We also provide the uncropped western blots images as source data (“raw-data WB” file).

**Author response image 4. sa3fig4:** In our previous study, we found that worms treated with vhp-1(RNAi), which hyperactivates p-PMK-1 (lower band) 4. In contrast, the two bands are disappeared in pmk-1 mutant. These pictures are extracted from our previous study4.

(3) Heat-killed *E. coli* (HK-*E. coli*) is low-quality because the lack of sugar cannot support the growth of *C. elegans* larvae (Qi and Han, Cell, 2018). Thus, animals do not show the UPRER-immune response and avoidance when HK-*E. coli* is supplemented with sugars such as glucose (Line 225-227). If these sugars are the key, *C. elegans* larvae should be able to grow better with HK-*E. coli* supplemented with glucose. Authors should address this possibility.

Previous studies have shown that heat-killed *E. coli* (HK-*E. coli*) is a low-quality food source that cannot support the growth of *C. elegans* larvae7. Here, we found that sugar deficiency in HK-*E. coli* induces the UPRER-immune response and avoidance behavior in *C. elegans*. Given this, we investigated whether sugar supplementation could promote animal growth when fed HK-*E. coli.* To our surprise, supplementing HK-*E. coli* with carbohydrates (D-Glc, D-GlcA) did not support animal development (Figure 3-figure supplement 2G), suggesting that carbohydrates are not essential for supporting animal growth on this food source. However, we did find that carbohydrates are critical for inhibiting the UPRER-immune response induced by sugar deficiency in HK-*E. coli.*

(4) Line 884: Instead of the Student's t-test, the ANOVA should be used for multiple comparisons.

All statistical analyses were performed in Graphpad prism 8.0. Two-tailed unpaired t test was used for statistical analysis of two groups of samples，one-way or two-way ANOVA was used for statistical analysis of more than two groups of samples.

(5) Although the results are interesting and convincing, the manuscript needs some careful editing and proofreading. As far as I could catch, there are more than 100 errors and typos, as I summarized in minor comments. I recommend the authors proofread thoroughly to make this work easier to read.

Thanks for the reviewer’s efforts. We changed all of these errors and polish the language of this paper.

Minor comments:

(1) Line 30: nature -> natural

(2) Line 86: elegnas -> elegans

(3) Line 93: the17h -> the 17h

(4) Line 97: response -> respond

(5) Line106: responded -> respond

(6) Lien 107-109: Add references for the three reporters

(7) Line 114: immune -> immune pathway

(8) Line 118: immune depended -> immune-dependent

(9) Line 128, 594, 596: deferentially -> differentially

(10) Line 131: Explain what IRE-1-mediated splicing of xbp-1 with references

(11) Line 170: XPB-1 -> XBP-1

(12) Line 179: URP -> UPR

(13) Line 181: hsp-4::GFP -> Phsp-4::GFP

(14) Line 183: Italicize *E. coli*; mutant -> mutants

(15) Line 184: irg-5::GFP -> Pirg-5::GFP (2 places)

(16) Line 197, 203, 206, 207: Lactose -> lactose

(17) Line 206, 209, 217, 225, 228, 232, 237, 262, 442, 445, 604, 739: Glucose -> glucose

(18) Line 218: Sugars deficiency -> sugar deficiency

(19) Line 229: found contribute to -> found to contribute to

(20) Line 235, 537, 539, 587, 599, 642, 855: Italicize *E. coli*

(21) Line 236: same -> the same

(22) Line 239: I recommend adding "in *C. elegans*". This study uses both *E. coli* and C.

elegans genetics. Sometimes, it is confusing which organism was mentioned. It should be applied where it is necessary.

(23) Line 240: additional -> addition

(24) Line 339, 642: Italicize kgb-1

(25) Line 390: Italicize *Pseudomonas aeruginosa*, Bacillus thuringiensis,

*Staphylococcus aureus*, and *Serratia marcescens*

(26) Line 394: wiht -> with

(27) Line 400, 550: Change ER to superscript; Italicize ire-1, xbp-1, and pmk-1

(28) Line 415: xpb-1 -> xbp-1

(29) Line 460, 525, 531, 532, 617, 655: Italicize yfbR

(30) Line 457, 468, 472, 475, 482, 497, 513, 624, 629, 633, 733. 758: Vitamin -> vitamin

(31) Line 459: Make it clear what is the relationship between vitamin C and TAA

(32) Line 527: Do not italicize mutant

(33) Line 538: Phsp-6:GFP -> Phsp-6::GFP (to match other descriptions)

(34) Line 540: Phsp-4:GFP -> Phsp-4::GFP (to match other descriptions)

(35) Line 540: Italicize hsp-4

(36) Line 543: Pirg-5:GFP -> Pirg-5::GFP (to match other descriptions) and italicize irg-5

(37) Line 550, 881: Innate -> innate

(38) Line 557, 560, 564, 838: Do not italicize HK

(39) Line 561: Remove the extra space before "three"

(40) Line 575, 577: Reporter -> reporter

(41) Line 575, 607: Italicize Phsp-4::GFP

(42) Line 577: immunity -> Immunity; Italicize Pirg-5::GFP

(43) Line 585, 653: keio -> Keio

(44) Line 586: hsp-4::GFP -> Phsp-4::GFP

(45) Line 586, 589 (2 places): irg-5::GFP -> Pirg-5::GFP

(46) Line 597: Remove "all"

(47) Line 600: Trehalose -> trehalose

(48) Line 609: Italicize Pirg-5::GFP

(49) Line 615: critically -> critical

(50) Line 636: Remove "+"

(51) Line 656 (2 places), 682: Do not italicize OP50

(52) Line 664: Lead -> lead

(53) Line 681: Describe the composition of NGM or show the reference. Since this paper examines nutrition, the composition of the medium is crucial.

(54) Line 686-706: Italicize all allele names. Be consistent with how to write the promoter to avoid confusion (e.g., ttx-3p -> Pttx-3). Be consistent with how to describe the transgene (e.g., Phsp-4::GFP(zcIs4) -> zcIs4[Phsp-4::GFP])

(55) Line 710: Describe the composition of LB or show the reference. Since this paper examines nutrition, the composition of the medium is crucial.

(56) Line 709, 856 (2 places), 858: Do not italicize K12 to make it consistent

(57) Line 719: Podr-1p:RFP -> Podr-1::RFP

(58) Line 722, 724: Italicize ges-1 and xbp-1

(59) Line 723: Pges-1:xbp-1::GFP -> Pges-1::xbp-1::GFP

(60) Line 735: Glucuronic -> glucuronic

(61) Line 748: I believe it is 5 mm instead of 0.5 mm

(62) Line 750: The equation should be (5 mm)2/(17.5 mm)2

(63) Line 759: Remove the period after "pattern".

(64) Line 766: Describe how they were synchronized

(65) Line 774: Italicize Psysm-1p::GFP

(66) Line 785: Insert a space before "until"

(67) Line 787: the mutant -> mutant

(68) Line 789, 792, 793, 795 (2 places): GPF -> GFP

(69) Line 791: next -> Next; an -> a

(70) Line 799: Remove a space before "MRC".

(71) Line 804: I do not understand what "until adulthood" means in this context;

Remove a space before "by". (I recommend searching double space and correcting it.)

(72) Line 853: Metabolome -> metabolome

(73) Line 893-1082: Species and gene names should be italicized in Reference

(74) Figures 1F, 1G, S2F, S2G: The panels' order should match the bar graphs' order. The apparent difference in the representative data does not match the marginal difference in the bar graph in Fig. 1G. The authors should double-check the results.

(75) Figure 1F, 2A, 2B, 3C, 3D, 3E, 4D, 4I, S1J, S2A, S2B, S2I, S3B, S3F, S3H: hsp-4::GFP -> Phsp-4::GFP

(76) Figure 1G, 2D, 3F, 4E, 4J, S1K, S2H, S3C, S3I: irg-5::GFP -> Pirg-5::GFP

(77) Figure 6: Liquids -> Lipids; Italicize ire-1, xbp-1, pmk-1

(78) Figure S1I: hsp-6::GFP -> Phsp-6::GFP

(79) In the legend for Figure S1 after Figure S1, (A), (B)... were duplicated. It is OK in the corresponding main text (Line 530)

(80) Figure S2F, S3G, S4C, S4D: sysm-1::GFP -> Psysm-1::GFP

(81) Figure S2G: irg-1::GFP -> Pirg-1::GFP

(82) Figure S3H and S3I: Describe which ones are Glu + conditions

References:

(1) Patananan AN, Budenholzer LM, Pedraza ME, Torres ER, Adler LN, Clarke SG. The invertebrate *Caenorhabditis elegans* biosynthesizes ascorbate. *Arch Biochem Biophys* 569, 32-44 (2015).

(2) Yabuta Y_, et al._ L-Ascorbate Biosynthesis Involves Carbon Skeleton Rearrangement in the Nematode *Caenorhabditis elegans*. *Metabolites* 10, (2020).

(3) Weaver BP, Weaver YM, Omi S, Yuan W, Ewbank JJ, Han M. Non-Canonical Caspase Activity Antagonizes p38 MAPK Stress-Priming Function to Support Development. *Dev Cell* 53, 358-369 e356 (2020).

(4) Geng S_, et al._ Gut commensal *E. coli* outer membrane proteins activate the host food digestive system through neural-immune communication. *Cell Host Microbe* 30, 1401-1416 e1408 (2022).

(5) Richardson CE, Kooistra T, Kim DH. An essential role for XBP-1 in host protection against immune activation in *C. elegans*. *Nature* 463, 1092-1095 (2010).

(6) Harding HP_, et al._ An Integrated Stress Response Regulates Amino Acid Metabolism and Resistance to Oxidative Stress. *Molecular Cell* 11, 619-633 (2003).

(7) Qi B, Kniazeva M, Han M. A vitamin-B2-sensing mechanism that regulates gut protease activity to impact animal’s food behavior and growth. *eLife* 6, e26243 (2017).

(8) Calfon M_, et al._ IRE1 couples endoplasmic reticulum load to secretory capacity by processing the XBP-1 mRNA. *Nature* 415, 92-96 (2002).

(9) Bolz DD, Tenor JL, Aballay A. A Conserved PMK-1/p38 MAPK Is Required in *Caenorhabditis elegans* Tissue-specific Immune Response to Yersinia pestis Infection*. *The Journal of Biological Chemistry* 285, 10832 - 10840 (2010).